# Mobile-Agent-v2: Mobile Device Operation Assistant with Effective Navigation via Multi-Agent Collaboration

**Junyang Wang**[1][*]    **Haiyang Xu**[2][†]    **Haitao Jia**[1]    **Xi Zhang**[2]    **Ming Yan**[2][†]
**Weizhou Shen**[2]    **Ji Zhang**[2]    **Fei Huang**[2]    **Jitao Sang**[1][†]

{junyangwang, 23120356, jtsang}@bjtu.edu.cn
{shuofeng.xhy, zx443053, ym119608, shenweizhou.swz, zj122146, f.huang}@alibaba-inc.com
[1]Beijing Jiaotong University    [2]Alibaba Group

## Abstract

Mobile device operation tasks are increasingly becoming a popular multi-modal AI application scenario. Current Multi-modal Large Language Models (MLLMs), constrained by their training data, lack the capability to function effectively as operation assistants. Instead, MLLM-based agents, which enhance capabilities through tool invocation, are gradually being applied to this scenario. However, the two major navigation challenges in mobile device operation tasks — task progress navigation and focus content navigation — are difficult to effectively solve under the single-agent architecture of existing work. This is due to the overly long token sequences and the interleaved text-image data format, which limit performance. To address these navigation challenges effectively, we propose Mobile-Agent-v2, a multi-agent architecture for mobile device operation assistance. The architecture comprises three agents: planning agent, decision agent, and reflection agent. The planning agent condenses lengthy, interleaved image-text history operations and screens summaries into a pure-text task progress, which is then passed on to the decision agent. This reduction in context length makes it easier for decision agent to navigate the task progress. To retain focus content, we design a memory unit that updates with task progress by decision agent. Additionally, to correct erroneous operations, the reflection agent observes the outcomes of each operation and handles any mistake accordingly. Experimental results indicate that Mobile-Agent-v2 achieves over a 30% improvement in task completion compared to the single-agent architecture of Mobile-Agent. The code is open-sourced at https://github.com/X-PLUG/MobileAgent.

## 1 Introduction

Multi-modal Large Language Models (MLLMs), represented by GPT-4v OpenAI (2023), have demonstrated outstanding capabilities in various domains Bai et al. (2023); Liu et al. (2023c,b); Dai et al. (2023); Zhu et al. (2023); Chen et al. (2023); Ye et al. (2023a,b); Wang et al. (2023c); Hu et al. (2023, 2024); Zhang et al. (2024b). With the rapid development of agents based on Large Language Models (LLMs) Zhao et al. (2024); Liu et al. (2023f); Talebirad and Nadiri (2023); Zhang et al. (2023b); Wu et al. (2023); Shen et al. (2024); Li et al. (2023a), MLLM-based agents, which can overcome the limitations of MLLMs in specific application scenarios by various visual perception tools, have become a focal point of research attention Liu et al. (2023d).

---

[*]Work done during internship at Alibaba Group.
[†]Corresponding author

38th Conference on Neural Information Processing Systems (NeurIPS 2024).

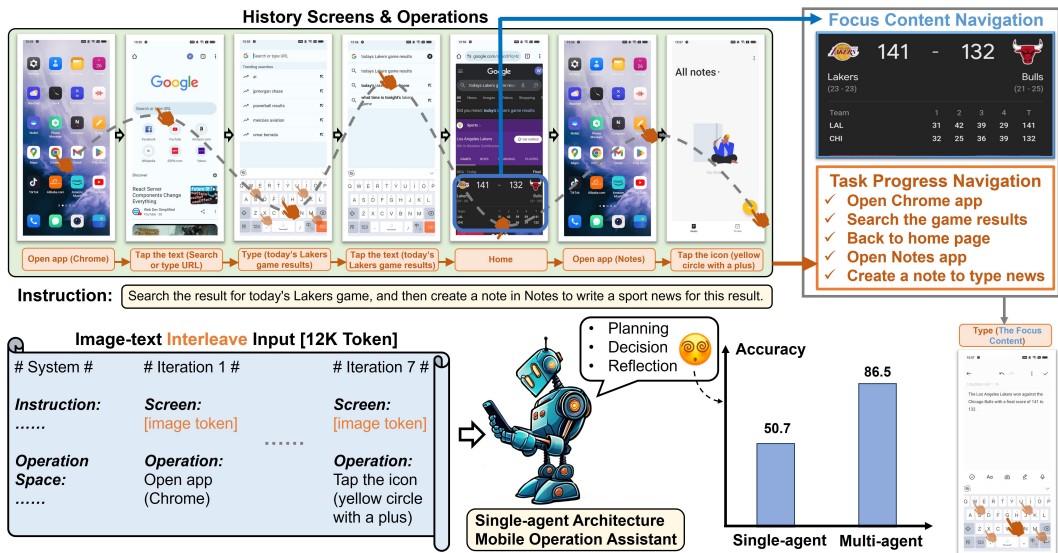

Figure 1: Mobile device operation tasks require navigating focus content and task progress from history operation sequences, where the focus content comes from previous screens. As the number of operations increases, the length of the input sequences grows, making it extremely challenging for a single-agent architecture to manage these two types of navigation effectively.

Automated operations on mobile devices, as a practical multi-modal application scenario, are emerging as a major technological revolution in AI smartphone development Yao et al. (2022); Deng et al. (2023); Gur et al. (2024); Zheng et al. (2024); Zhang et al. (2023a); Wang et al. (2024); Chen and Li (2024a,b,c); Zhang et al. (2024a); Wen et al. (2023); Cheng et al. (2024). However, due to the limited screen recognition, operation, and location capabilities, existing MLLMs face challenges in this scenario. To address this, existing work leverages MLLM-based agent architecture to endow MLLMs with various capabilities for perceiving and operating mobile device UI. AppAgent Zhang et al. (2023a) tackles the limitation of MLLMs in localization by extracting clickable positions from device XML files. However, the reliance on UI files limits the applicability of this method to other platforms and devices. To eliminate the dependency on underlying UI files, Mobile-Agent Wang et al. (2024) proposes a solution for localization through visual perception tools. It perceives the screen through an MLLM and generates operations, locating their positions by visual perception tools.

Mobile device operation tasks involve multi-step sequential processing. The operator needs to perform a series of continuous operations on the device starting from the initial screen until the instructions are fully executed. There are two main challenges in this process. First, to plan the operation intent, the operator needs to navigate the current task progress from the history operations. Second, some operations may require task-relevant information in the history screens, for example, writing sports news in Figure 1 requires using the match results queried earlier. We refer to this important information as the focus content. The focus content also needs to be navigated out from the history screens. However, as the task progresses, the lengthy history of interleaved image and text history operations and screens as input can significantly reduce the effectiveness of navigation in a single-agent architecture, as shown in Figure 1.

In this paper, we propose Mobile-Agent-v2, a mobile device operation assistant with effective navigation via multi-agent collaboration. Mobile-Agent-v2 has three specialized agent roles: **planning agent**, **decision agent**, and **reflection agent**. The planning agent needs to generate a task progress based on the history operations. To save the focus content from the history screens, we design a **memory unit** to record task-related focus content. This unit will be observed by the decision agent when generating an operation, simultaneously checking if there is any focus content on the screen and updating it to the memory. Since the decision agent cannot observe the previous screen to reflect, we design the reflection agent to observe the changes in the screen before and after the decision agent's operation and determine whether the operation meets the expectations. If it finds that the operation does not meet expectations, it will take appropriate measures to re-execute the operation. The entire

process is illustrated in Figure 3. The three agent roles work respectively in the progress, decision, and reflection stages, collaborating to alleviate the difficulty of navigating.

Our summarized contributions are as follows:

- We propose a multi-agent architecture Mobile-Agent-v2 to alleviate various navigating difficulties inherent in the single-agent framework for mobile device operation tasks. We design a planning agent to generate task progress based on the history operations, ensuring effective operation generation by the decision agent.

- To avoid the loss of focus content navigating and reflection capability, we design both a memory unit and a reflection agent. The memory unit is updated by the decision agent with focus content. The reflection agent assesses whether the decision agent's operation meets expectations and generates appropriate remedial measures if expectations are not met.

- We conducted dynamic evaluations of Mobile-Agent-v2 across various operating systems, language environments, and applications. Experimental results demonstrate that Mobile-Agent-v2 achieves significant performance improvements. Furthermore, we empirically validated that the performance of Mobile-Agent-v2 can be further enhanced by manual operation knowledge injection.

## 2 Related Work

### 2.1 Muiti-agent Application

The powerful comprehension and reasoning capabilities of Large Language Models (LLMs) enable LLM-based agents to demonstrate the ability to independently execute tasks Brown et al. (2020); Achiam et al. (2023); Touvron et al. (2023a,b); Bai et al. (2023). Inspired by human-team collaboration, the multi-agent framework has been proposed. Park et al. (2023) constructs Smallville consisting of 25 agents in a sandbox environment. Li et al. (2023b) proposes a role-playing-based multi-agent collaborative framework to enable two agents playing different roles to autonomously collaborate. Chen et al. (2024) innovatively propose an effective multi-agent framework for coordinating the collaboration of multiple expert agents. Hong et al. (2024) presents a groundbreaking meta-programming multi-agent collaboration framework. Wu et al. (2024) proposes a generic multi-agent framework that allows users to configure the number of agents, interaction modes, and toolsets. Chan et al. (2024); Subramaniam et al. (2024); Tao et al. (2024) investigate the implementation of a multi-agent debating framework, aiming to evaluate the quality of different texts or generated content. Abdelnabi et al. (2024); Xu et al. (2024); Mukobi et al. (2024) integrate multi-agent interaction with game theoretic strategies, aiming to enhance both the cooperative and decision abilities.

### 2.2 LLM-based UI Operation Agent

Webpages, as a classic application scenario for UI agents, have attracted widespread attention to research on web agents. Yao et al. (2022) and Deng et al. (2023) aim to enhance the performance of agents on real-world webpage tasks by constructing high-quality website task datasets. Gur et al. (2024) utilizes pre-trained LLMs and self-experience learning to automate task processing on real-world websites. Zheng et al. (2024) leverages GPT-4V for visual understanding and webpage manipulation. Simultaneously, research on LLM-based UI agents for mobile platforms has also drawn significant attention. Wen et al. (2023) converts Graphical User Interface (GUI) information into HTML representations and then leverages LLM in conjunction with application-specific domain knowledge. Yan et al. (2023) proposes a multi-modal intelligent mobile agent based on GPT-4V, exploring the direct utilization of GPT-4V to perceive screen screenshots with annotations. Unlike the former approach that operates on screens with digital labels, Zhang et al. (2023a) combines the application's XML files for localization operations, mimicking human spatial autonomy in operating mobile applications. Wang et al. (2024) eliminates the dependency on the application's XML files and leverages visual module tools for localization operations. Additionally, Hong et al. (2023) designed a GUI agent based on pre-trained vision-language models. Chen and Li (2024a,b,c) propose small-scale client-side models for deployment on actual devices. Zhang et al. (2024a) proposed a UI multi-agent framework tailored for the Windows operating system. Despite the significant performance improvements achieved by multi-agent architectures in many tasks, currently, there is no work that employs multi-agent architectures in mobile device operation tasks. To address the

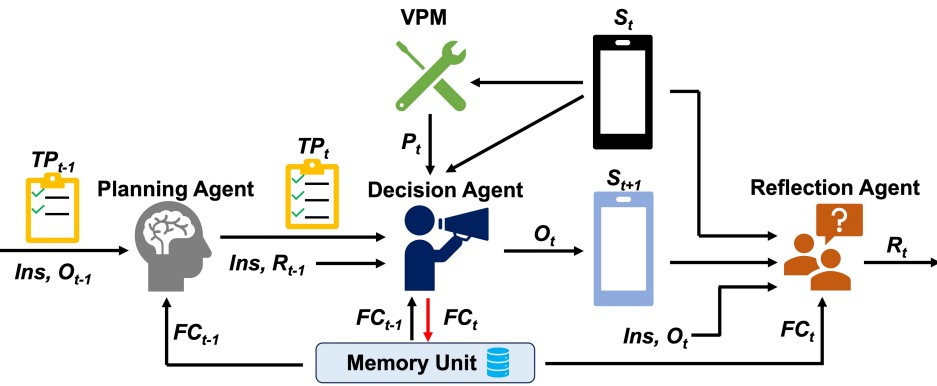

Figure 2: Illustration of the overall framework of Mobile-Agent-v2.

challenges of long-context navigation in mobile device operation tasks, in this paper, we introduce the multi-agent architecture Mobile-Agent-v2.

# 3 Mobile-Agent-v2

In this section, we will provide a detailed overview of the architecture of Mobile-Agent-v2. The operation of Mobile-Agent-v2 is iterative, and its process is depicted in Figure 2. Mobile-Agent-v2 has three specialized agent roles: planning agent, decision agent, and reflection agent. We also design the visual perception module and memory unit to enhance the agent's screen recognition capability and the capability to navigate focus content from history. Firstly, the planning agent updates the task progress, allowing the decision agent to navigate the progress of the current task. The decision agent then operates based on the current task progress, current screen state, and the reflection (if the last operation is erroneous). Subsequently, the reflection agent observes the screens before and after the operation to determine if the operation meets expectations.

## 3.1 Visual Perception Module

Screen recognition remains challenging even for state-of-the-art MLLMs when processed end-to-end. Therefore, we have incorporated a visual perception module to enhance the screen recognition capability. In this module, we utilize three tools: text recognition tool, icon recognition tool, and icon description. Inputting a screenshot into this module will ultimately yield the text and icon information present on the screen, along with their respective coordinates. This process is represented by the following formula:

$$P_t = VPM(S_t) \tag{1}$$

where $P_t$ represents the perception result of the screen in the $t$-th iteration.

## 3.2 Memory Unit

Due to the task progress generated by the planning agent being in textual form, the navigation of focus content from history screens is still challenging. To address this issue, we design a memory unit to store the focus content related to the current task from history screens. The memory unit serves as a short-term memory module that is updated as the task progresses. The memory unit is crucial for scenarios involving multiple apps. For instance, as shown in Figure 3, weather information observed by the decision agent will be utilized in subsequent operations. At this point, the information related to the weather app's page will be updated in the memory unit.

## 3.3 Planning Agent

We aim to reduce the reliance on lengthy history operations during decision-making by employing a separate agent. We observe that although each round of operation occurs on different pages and is different, often the goals of multiple operations are the same. For example, in the example illustrated

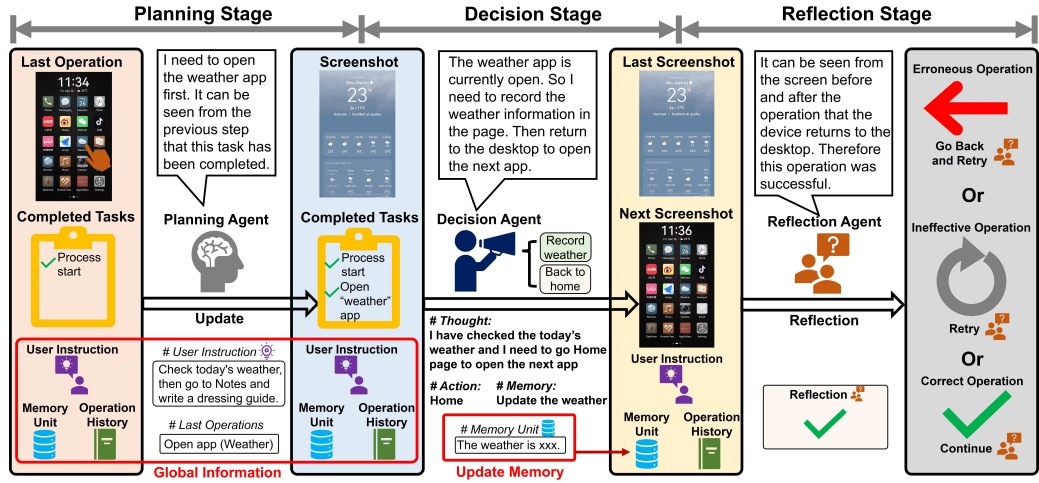

Figure 3: Illustration of the operation process and interaction of agent roles in Mobile-Agent-v2.

in Figure 1, the first four operations are all about searching for match results. Therefore, we design a planning agent to summarize the history operations and track task progress.

We define the operation generated by the decision agent at the $t$-th iteration as $O_t$. Before the decision agent makes a decision, the planning agent observes the decision agent's operation $O_{t-1}$ from the last iteration and updates the task progress $TP_{t-1}$ to $TP_t$. The task progress includes the sub-tasks that have already been completed. After generating the task progress, the planning agent passes it to the decision agent. This aids the decision agent in considering the content of tasks that have not yet been completed, thereby facilitating the generation of the next operation. As shown in Figure 3, the planning agent's inputs consist of four parts: user instruction $Ins$, the focus content $FC_t$ in memory unit, the previous operation $O_{t-1}$, and the previous task progress $TP_{t-1}$. Based on above information, the planning agent generates the $TP_t$. This process is represented by the following formula:

$$TP_t = PA(Ins, O_{t-1}, TP_{t-1}, FC_{t-1}) \tag{2}$$

where the $PA$ represents the LLM of the planning agent.

### 3.4 Decision Agent

The decision agent operates during the decision stage, generating operations $O$ and implementing them on the device, while also being responsible for updating the focus content $FC$ in memory unit. This process is illustrated in the Decision Stage shown in Figure 3 and represented by the following formula:

$$O_t = DA(Ins, TP_{t-1}, FC_{t-1}, R_{t-1}, S_t, P_t) \tag{3}$$

where the $DA$ represents the MLLM of the decision agent and the $R_t$ represents the reflection result from reflection agent.

**Operation Space.** To reduce the complexity of operations, we design an operation space and restricted the decision agent to selecting operations only from within this space. For operations with higher degrees of freedom, such as tapping and swiping, we incorporate an additional parameter space to locate or handle specific content. Below is a detailed description of the operation space:

- Open app (*app name*). If the current page is the home page, this operation can be used to open the app named "*app name*".

- Tap (*x*, *y*). This operation is used to tap on the position with coordinates (*x*, *y*).

- Swipe (*x1*, *y1*), (*x2*, *y2*). This operation is used to swipe from the position with coordinates (*x1*, *y1*) to the position with coordinates (*x2*, *y2*).

- Type (*text*). If the current keyboard is in an active state, this operation can be used to input the content of "*text*" in the input box.

- Home. This operation is used to return to the home page from any page.
- Stop. If the decision agent thinks that all requirements have been fulfilled, it can use this operation to terminate the entire operation process.

**Memory Unit Update.** As each operation made by the decision agent is highly task-relevant and based on the visual perception results of the current page, it is well-suited to observe task-related focus content within the screen pages. Accordingly, we have endowed the decision agent with the ability to update the memory unit. When making decisions, the decision agent is prompted to observe whether there is task-related focus content within the current screen page. If such information is observed, the decision agent updates it in memory for reference in subsequent decisions. This process is represented by the following formula:

$$FC_t = DA(Ins, FC_{t-1}, S_t, P_t) \qquad (4)$$

### 3.5 Reflection Agent

Even with the visual perception module, Mobile-Agent-v2 may still generate unexpected operations. In some specific scenarios, MLLMs may even produce severe hallucinations Liu et al. (2023a); Li et al. (2023c); Gunjal et al. (2024); Wang et al. (2023b); Zhou et al. (2023), even the most advanced MLLM GPT-4V Cui et al. (2023); Wang et al. (2023a). Therefore, we design the reflection agent to observe the screen state before and after a decision agent's operation to determine whether the current operation meets expectations. This process is represented by the following formula:

$$R_t = RA(Ins, FC_t, O_t, S_t, P_t, S_{t+1}, P_{t+1}) \qquad (5)$$

where the $RA$ represents the MLLM of the reflection agent.

As shown in Figure 3, the reflection agent generates three types of reflection results after operation execution: erroneous operation, ineffective operation, and correct operation. The following will describe these three reflection results:

- Erroneous operation refers to an operation that leads the device to enter a page unrelated to the task. For example, the agent intends to chat with contact $A$ in a messaging app, but it accidentally opens the chat page of contact $B$ instead.
- Ineffective operation refers to an operation that does not result in any changes to the current page. For example, the agent intends to tap on an icon, but it taps on the blank space next to the icon instead.
- Correct operation refers to an operation that meets the decision agent's expectations and serves as a step towards fulfilling the requirements of the user instruction.

If erroneous operation, the page will revert to the state before the operation. If ineffective operation, the page will remain in its current state. Neither erroneous nor ineffective operations are recorded in the operation history to prevent the agent from following these operations. If correct operation, the operation will be updated in the operation history, and the page will be updated to the current state.

## 4 Experiments

### 4.1 Model

**Visual Perception Module.** For the text recognition tool, we use the document OCR recognition model ConvNextViT-document from ModelScope[3]. For the icon recognition tool, we employ GroundingDINO Liu et al. (2023e), a detection model capable of detecting objects based on natural language prompts. For the icon description tool, we utilize the Qwen-VL-Int4[4].

**MLLMs.** For the planning agent, as it does not require screen perception, we utilize the text-only GPT-4 OpenAI (2023). For the decision agent and reflection agent, we employ GPT-4V OpenAI (2023). All calls are made through the official API method provided by the developers.

---

[3]https://modelscope.cn/models/iic/cv_convnextTiny_ocr-recognition-document_damo/summary

[4]https://modelscope.cn/models/qwen/Qwen-VL-Chat-Int4/summary

## 4.2 Evaluation

**Evaluation Method.** To evaluate the performance of Mobile-Agent-v2 on real mobile devices, we employed a dynamic evaluation method. This evaluation method requires the operation tool to implement the agent's operations in real time on an actual device. We use two mobile operation systems, Harmony OS and Android OS, to assess capabilities in non-English and English scenarios, respectively. We used Android Debug Bridge (ADB) as the tool for operating mobile devices. ADB can simulate all operations of Mobile-Agent-v2 in the operation space. In both scenarios, we select 5 system apps and 5 popular external apps for evaluation. For each app, we devise two basic instructions and two advanced instructions. Basic instructions are relatively simple operations with clear instructions within the app interface, while advanced instructions require a certain level of experience with app operations to complete. Additionally, to evaluate multi-app operation capability, we design two basic instructions and two advanced instructions involving multiple apps. In total, there were 88 instructions for non-English and English scenarios, comprising 40 instructions for system apps, 40 instructions for external apps, and 8 instructions for multi-app operations.[5] The apps and instructions used for evaluation in non-English and English scenarios are presented in the appendix.

**Metrics.** We design the following four metrics for dynamic evaluation:

- Success Rate (SR): When all the requirements of a user instruction are fulfilled, the agent is considered to have successfully executed this instruction. The success rate refers to the proportion of user instructions that are successfully executed.

- Completion Rate (CR): Although some challenging instructions may not be successfully executed, the correct operations performed by the agent are still noteworthy. The completion rate refers to the proportion of correct steps out of the ground truth operations.

- Decision Accuracy (DA): This metric reflects the accuracy of the decision by the decision agent. It is the proportion of correct decisions out of all decisions.

- Reflection Accuracy (RA): This metric reflects the accuracy of reflection by the reflection agent. It is the proportion of correct reflections out of all reflections.

**Implementation Details.** We use Mobile-Agent as the baseline. Mobile-Agent is a single-agent architecture based on GPT-4V end-to-end screen recognition. We fix the seed for GPT-4V invocation and set the temperature to 0 to avoid randomness. In addition to Mobile-Agent-v2, we further introduce the scenario of knowledge injection. This involves providing the agent with some operation hints in addition to the user instructions to aid the agent. It's worth noting that we only injected knowledge for instructions that Mobile-Agent-v2 couldn't accomplish. For instructions that could be completed without additional assistance, we keep the input unchanged.

## 4.3 Results

### 4.3.1 Evaluation

**Evaluation on Task Completion.** Table 1 and 2 respectively illustrate the performance of Mobile-Agent-v2 in non-English and English scenarios. Compared to Mobile-Agent, Mobile-Agent-v2 exhibits significant improvements in both basic and advanced instructions. With the multi-agent architecture, even in highly challenging advanced instructions, the success rate can still reach 55%, compared to only 20% with Mobile-Agent. Even in the English scenario, Mobile-Agent-v2 still achieves a significant performance improvement. Although Mobile-Agent performs better in English scenarios compared to Chinese scenarios, Mobile-Agent-v2 still achieves an average improvement of 27% in success rate.

**Evaluation on Reflection Capability.** In the case of knowledge injection, even though the decision accuracy does not reach 100%, the completion rate can still reach 100%. This indicates that even with knowledge injection, Mobile-Agent-v2 still makes erroneous decisions. Errors in decision-making are difficult to avoid, even for humans. Therefore, the importance of the reflection agent is highlighted.

---

[5]We did not use Mobile-Eval because the difficulty of this benchmark is relatively low and Mobile-Agent-v2 can achieve an accuracy of 99%. In this work, we have redesigned more challenging evaluation tasks.

| Method | Basic Instruction | | | | Advanced Instruction | | | |
|---|---|---|---|---|---|---|---|---|
| | SR | CR | DA | RA | SR | CR | DA | RA |
| | *System app* | | | | | | | |
| Mobile-Agent | 5/10 | 41.2 | 37.6 | - | 3/10 | 37.3 | 32.9 | - |
| Mobile-Agent-v2 | 9/10 | 86.8 | 82.5 | 93.3 | 6/10 | 82.7 | 78.2 | 84.4 |
| Mobile-Agent-v2 + *Know.* | 10/10 | 100 | 98.2 | 98.9 | 8/10 | 88.9 | 87.2 | 91.4 |
| | *External app* | | | | | | | |
| Mobile-Agent | 2/10 | 38.3 | 35.4 | - | 1/10 | 29.2 | 27.0 | - |
| Mobile-Agent-v2 | 8/10 | 97.9 | 94.0 | 92.5 | 5/10 | 77.9 | 74.1 | 78.8 |
| Mobile-Agent-v2 + *Know.* | 10/10 | 100 | 95.6 | 97.3 | 8/10 | 87.8 | 83.0 | 85.9 |
| | *Multi-app* | | | | | | | |
| Mobile-Agent | 1/2 | 52.8 | 50.0 | - | 0/2 | 33.3 | 31.4 | - |
| Mobile-Agent-v2 | 2/2 | 100 | 92.9 | 91.6 | 2/2 | 100 | 93.8 | 92.9 |
| Mobile-Agent-v2 + *Know.* | - | - | - | - | - | - | - | - |

Table 1: Dynamic evaluation results on non-English scenario, where the *Know.* represents manually injected operation knowledge.

| Method | Basic Instruction | | | | Advanced Instruction | | | |
|---|---|---|---|---|---|---|---|---|
| | SR | CR | DA | RA | SR | CR | DA | RA |
| | *System app* | | | | | | | |
| Mobile-Agent | 9/10 | 92.5 | 89.7 | - | 4/10 | 62.0 | 71.3 | - |
| Mobile-Agent-v2 | 9/10 | 95.0 | 92.9 | 96.5 | 6/10 | 76.0 | 77.6 | 88.4 |
| Mobile-Agent-v2 + *Know.* | 10/10 | 100 | 96.2 | 98.7 | 8/10 | 85.3 | 87.9 | 92.0 |
| | *External app* | | | | | | | |
| Mobile-Agent | 7/10 | 79.7 | 72.0 | - | 3/10 | 45.3 | 38.7 | - |
| Mobile-Agent-v2 | 9/10 | 97.1 | 93.8 | 96.2 | 7/10 | 89.7 | 91.0 | 93.4 |
| Mobile-Agent-v2 + *Know.* | 10/10 | 100 | 98.2 | 97.4 | 9/10 | 97.1 | 94.2 | 98.5 |
| | *Multi-app* | | | | | | | |
| Mobile-Agent | 2/2 | 100 | 91.2 | - | 1/2 | 86.7 | 92.9 | - |
| Mobile-Agent-v2 | 2/2 | 100 | 97.4 | 100 | 1/2 | 93.3 | 93.3 | 80.0 |
| Mobile-Agent-v2 + *Know.* | - | - | - | - | 2/2 | 100 | 100 | 100 |

Table 2: Dynamic evaluation results on English scenario, where the *Know.* represents manually injected operation knowledge.

**Evaluation on App Type.** From all metrics, it can be observed that the performance of all methods on system apps exceeds that of external apps. From the results of multiple apps, it can be seen that Mobile-Agent-v2 achieves improvements of 37.5% and 44.2% in SR and CR, respectively, compared to Mobile-Agent. Compared to single-app tasks, multi-app tasks rely more on the retrieval of history operations and focus content. The significant performance improvement indicates that the multi-agent architecture and memory unit of Mobile-Agent-v2 play an important role.

**Evaluation on Operation Knowledge Injection.** From the results of knowledge injection in Table 1 and 2, it can be observed that operation knowledge can effectively enhance the performance of Mobile-Agent-v2, which suggests that manually injected operation knowledge can mitigate the limitations of an agent's operation capability. This finding implies that knowledge injection can broaden the application scenarios of Mobile-Agent-v2 because even complex tasks can be guided by manually written operation tutorials to instruct agents. This finding may offer new insights for automated script testing on mobile devices and suggests that to enhance the operation capabilities of MLLMs to their limits, automating the generation of high-quality operation knowledge can further improve the performance of Mobile-Agent-v2. Moreover, the success brought about by knowledge injection also opens up new avenues for future mobile app testing. Existing mobile app testing solutions are still limited to manual script writing, which restricts the universality of testing and raises

| Model | Basic | Advanced |
|---|---|---|
| | SR&DA | SR&DA |
| GPT-4V w/o agent | 2.7 | 0.9 |
| Gemini-1.5-Pro | 38.2 | 29.8 |
| Qwen-VL-Max | 42.1 | 33.6 |
| GPT-4V | **92.7** | **83.5** |

Table 3: Performance results of Mobile-Agent-v2 with different MLLMs. To better illustrate the differences, we converted all instructions to single-step forms and evaluated the success rate (which is the same as decision accuracy) of each single-step task.

| Ablation Setting | | | Basic | | | Advanced | | |
|---|---|---|---|---|---|---|---|---|
| Planning Agent | Reflection Agent | Memory Unit | SR | CR | DA | SR | CR | DA |
| | ✓ | ✓ | 59.1 | 63.7 | 58.9 | 29.5 | 43.8 | 42.6 |
| ✓ | ✓ | | 77.3 | 83.6 | 84.0 | 45.5 | 72.3 | 69.8 |
| ✓ | | ✓ | 86.4 | 89.2 | 85.7 | 54.5 | 75.9 | 72.4 |
| ✓ | ✓ | ✓ | **88.6** | **93.9** | **89.4** | **61.4** | **82.1** | **80.3** |

Table 4: The results of the ablation study on planning agent, reflection agent, and memory unit.

the threshold for users. To address the aforementioned issues, one can inject natural language testing procedures into Mobile-Agent-v2. After injecting accurate testing procedures, the system can operate normally regardless of changes in the size or color of the mobile interface. Additionally, language descriptions eliminate the need for a knowledge base in script writing.

**Evaluation on MLLMs.** In Table 3, we evaluate the performance of different MLLMs within the Mobile-Agent-v2 framework. Since some models are not well-suited for handling sequential inputs, we selected specific instructions and modified each step to function as a single-step task. Therefore, we only evaluate the SR (which is the same as DA). We also evaluate the direct use of GPT-4V, bypassing the agent architecture for end-to-end operation. The results indicate that using GPT-4V directly as a mobile device operation assistant is almost infeasible. GPT-4V combined with the agent architecture remains the most effective configuration for operational capabilities.

### 4.3.2 Ablation Study

We conduct ablation studies on Mobile-Agent-v2, including the planning agent, reflection agent, and memory unit. From the results in Table 4, it is evident that the impact of the planning agent on the overall framework is the most significant. This further demonstrates the challenging nature of navigation in long sequences for current MLLMs. Additionally, performance declines are observed after removing the reflection agent and memory unit. The reflection agent is essential for correcting erroneous operations. It enables the decision agent to avoid operating on incorrect pages or getting stuck in loops of invalid operations. The memory unit is crucial for successful execution in multi-app scenarios. Even in scenarios involving multiple sub-tasks, the memory can sometimes record the positions of critical UI elements, aiding better localization for the next sub-task execution.

### 4.3.3 Analysis of Operation Sequence Length

As shown in Figure 4, we analyze the positions of erroneous or ineffective operations within failed instructions in English scenarios, dividing the relative sequence positions into three equal parts. The results indicate that in Mobile-Agent, such errors or ineffective operations predominantly occur in the later stages of the tasks. In contrast, Mobile-Agent-v2 does not exhibit any obvious pattern. This indicates that the multi-agent architecture is better equipped to handle the challenges posed by long sequences in UI operation tasks.

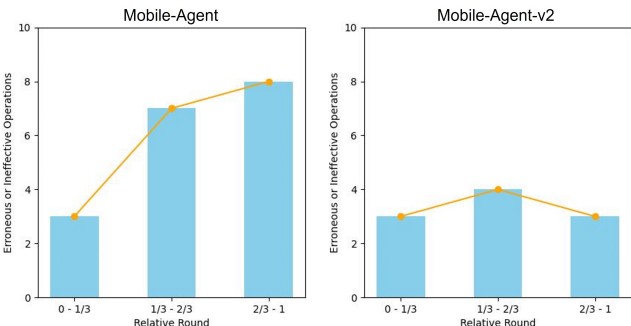

Figure 4: The relative positions of erroneous or ineffective operations in the operation sequence.

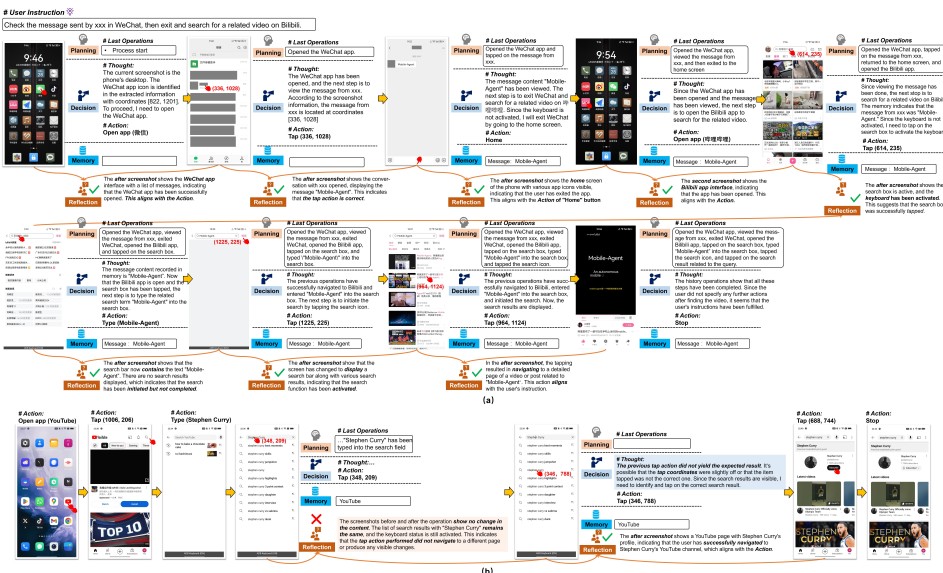

Figure 5: The cases of the complete operation process and reflection of Mobile-Agent-v2.

### 4.3.4 Case Study

Figure 5 (a) illustrates a complete operational process of Mobile-Agent-v2. Under the planning of the planning agent, the decision agent can navigate the task progress correctly in the case of single-image input. Meanwhile, the memory unit accurately stores the chat content needed for the task and, when required for search purposes, the decision agent can effectively navigate to it. Figure 5 (b) illustrates an example of correcting ineffective operations through reflection. After the failure of the previous operation, the reflection agent promptly detects the error and communicates the reflection result to the decision agent. Based on this, the decision agent rethinks and implements the correct operation.

## 5 Conclusion

Existing single-agent architectures for mobile device operation assistants experience a significant reduction in navigation effectiveness when dealing with long sequences of interleaved text and images, thereby limiting their performance. To address this issue, in this paper, we propose Mobile-Agent-v2, a mobile device operation assistant with efficient navigation via multi-agent collaboration. We address the aforementioned navigating challenges through the planning agent and memory unit, respectively. Additionally, we design reflectors to ensure the smooth progress of tasks. Experimental results demonstrate that Mobile-Agent-v2 achieves significant performance improvements compared to the single-agent Mobile-Agent. Furthermore, we find that performance can be further enhanced through the injection of manual operation knowledge, providing new directions for future work.

# 6 Acknowledgements

This work is supported by the National Key R&D Program of China (No. 2023YFC3310700).

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

# A    Appendix / supplemental material

## A.1    Evaluation Application and Instruction

Table 5 and 6 present the applications and instructions used for dynamic evaluation in both English and non-English scenarios. Basic instructions are relatively simple operations with clear instructions within the app interface, while advanced instructions require a certain level of experience with app operations to complete.

## A.2    Agent Prompt

In Table 7, 8, 9, and 10, we present the system and user prompts for the planning agent, decision agent, and reflection agent. Notably, since the history of operations is empty at the beginning of the task, we display the initial planning and subsequent planning prompts for the planning agent separately in Table 7 and 8.

For image inputs, we differentiated based on the task characteristics of each agent. For the planning agent, as navigating historical operations and generating task plans is a purely textual process, no screenshots are required. For the decision agent, we input the screenshot of the current device state. For the reflection agent, we additionally retain the screenshot from the previous operation and input it alongside the current device state screenshot in chronological order.

Since some instructions need to be specially tailored for specific mobile applications, we have hidden these details to protect privacy. In the Table 5 and 6, "xxx" represents the redacted information. For certain celebrities or locations mentioned in the instructions, we retained them as they do not involve privacy concerns.

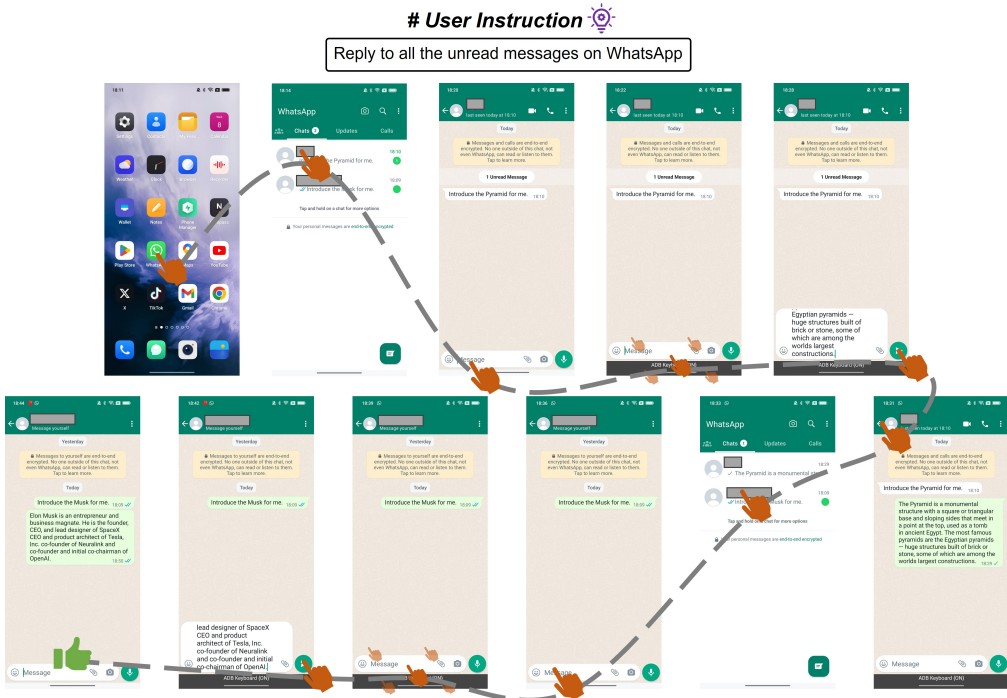

Figure 6: A case of replying to messages in chat platform WhatsApp based on the content of unread messages.

# User Instruction

Search for Elon Musk on X app and follow him

Figure 7: A case of searching for a celebrity on social media platform X and following him.

# User Instruction

Search for "Musk" on TikTok and open the relevant videos, then comment on the relevant content

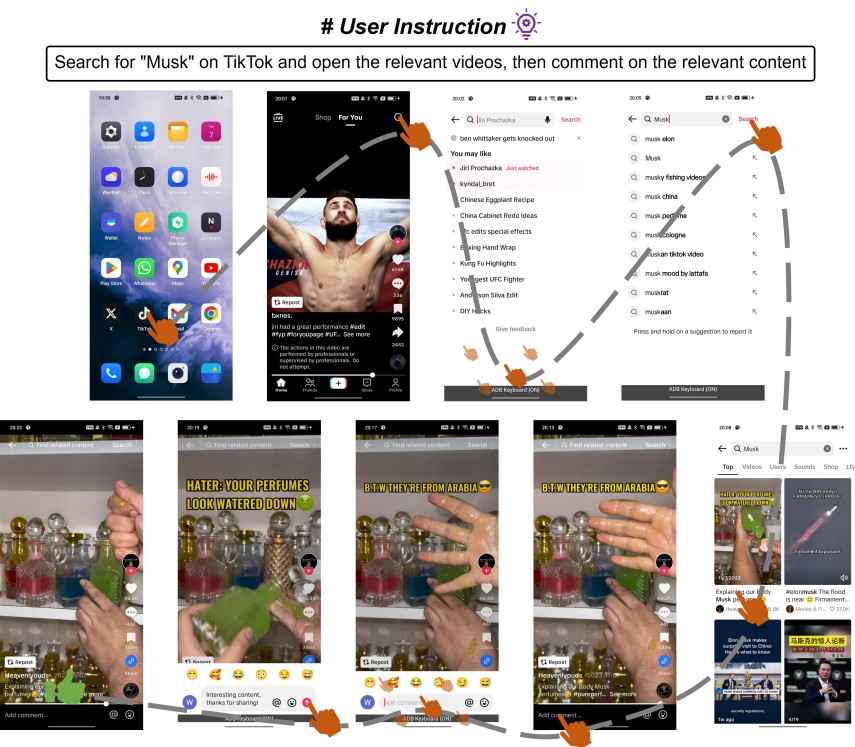

Figure 8: A case of searching for a celebrity's video on the short video platform TikTok and commenting on relevant content.

# *User Instruction* 💡

在小红书中搜索一个机器学习相关的帖子并评论相关内容

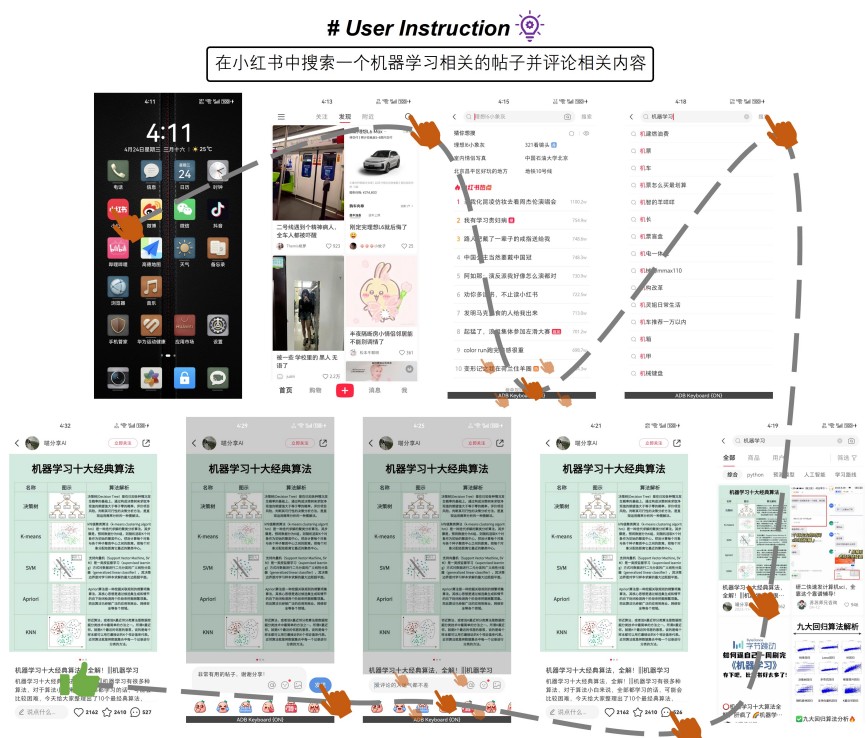

Figure 9: A case of searching for posts with specific content on the Little Red Book.

# *User Instruction* 💡

根据帖子内容评论微博首页中的一个帖子

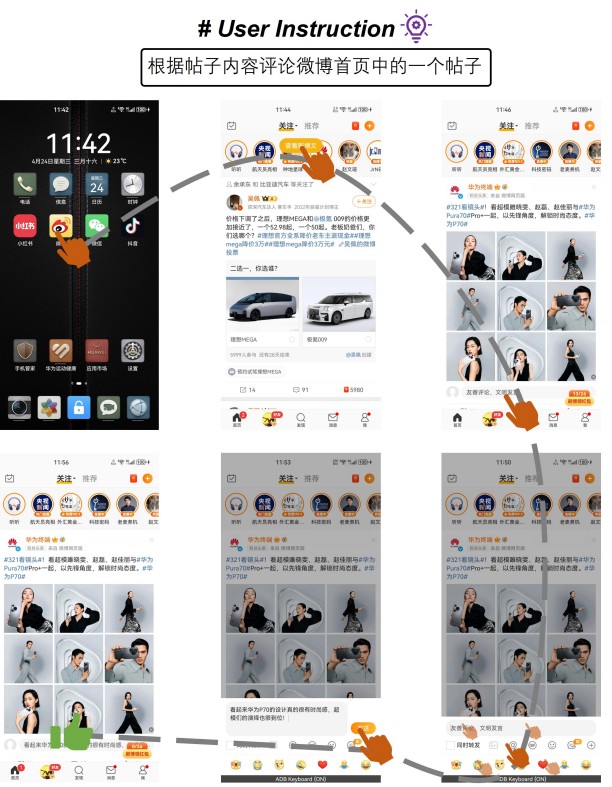

Figure 10: A case of commenting on a post on the social media platform Weibo.

# *User Instruction* 💡

根据帖子内容评论微博首页中的一个帖子

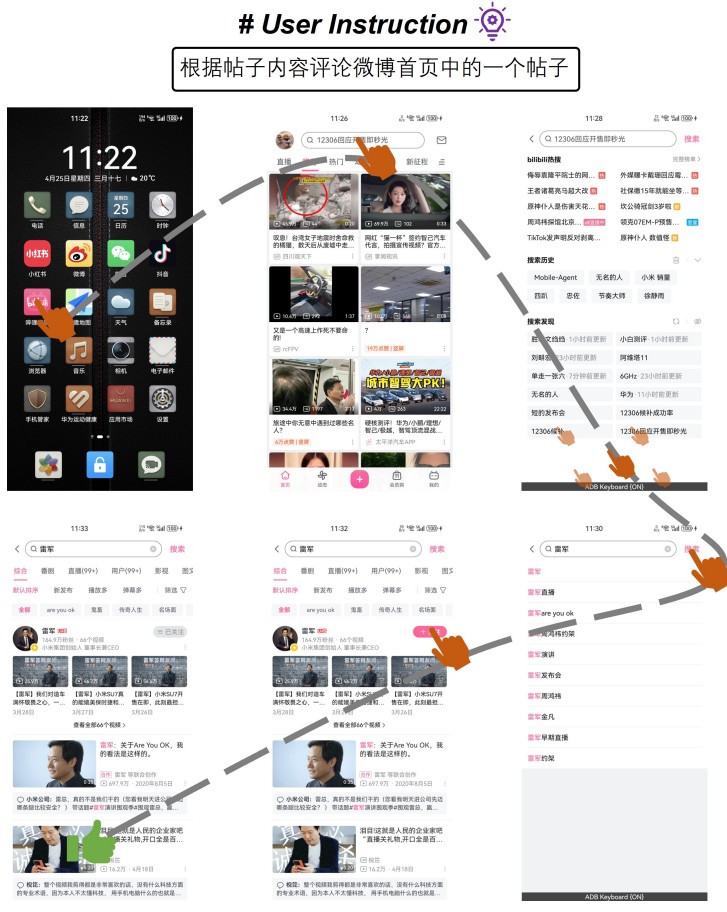

Figure 11: A case of searching for a celebrity and following them on the long-video platform Bilibili.

| App | Basic Instruction | Advanced Instruction |
|---|---|---|
| *System app* | | |
| Setting | 1. Turn on dark mode. 
 2. Set the system sound to "Do Not Disturb". | 1. Switch the system theme. 
 2. Turn on the real-time network speed display in the notification bar. |
| Cinema | 1. Open the camera and take a photo. 
 2. Open the camera and record a video. | 1. Open the camera and take a photo with a telephoto lens, and then view the photo. 
 2. Open the camera and record a video with a telephoto lens, and then view the video. |
| Contact | 1. Call the user in Phone with the phone number "123". 
 2. Reply to the unread text message according to the content of the text message. | 1. Call the xxx from the Contacts app. 
 2. Send a greeting message to xxx. |
| Notes | 1. Create a new note and write something. 
 2. Create a new note to record the current phone power and network speed. | 1. Delete the existing note in the note, then create a new note to record the current phone power and network speed and save it. 
 2. Create a new note, write something and return to the previous page, then create another note to record the current phone power and network speed. |
| System app | 1. Install "支付宝" in the App Market. 
 2. Create a new note to record the current phone power and network speed. | 1. Delete the existing note in the note, then create a new note to record the current phone power and network speed and save it. 
 2. Create a new note, write something and return to the previous page, then create another note to record the current phone power and network speed. |
| *External app* | | |
| X | 1. Like a post on the homepage of X app. 
 2. Search for Elon Musk on X app and follow him. | 1. Comment on a post on the homepage of X app with relevant content. 
 2. Search for Elon Musk on X app and comment on a post of him. |
| TikTok | 1. Swipe up on TikTok to see a cat-related video and like it. 
 2. Swipe up on TikTok to see a cat-related video and share it with other users. | 1. Swipe up on TikTok to see a cat-related video and comment on the relevant content. 
 2. Search for "Musk" on TikTok and open the relevant videos, then comment on the relevant content. |
| YouTube | 1. Like a video in "Shorts" on YouTube. 
 2. Search for "Stephen Curry" on YouTube and subscribe him. | 1. Like a Shorts video on YouTube and then comment on the relevant content. 
 2. Search for a video about "LeBron James", then like it and comment on the relevant content. |
| Google | 1. Navigate to gas station on Maps. 
 2. Install Facebook on Play store. | 1. Write an email and send it to "abc@cba.com". 
 2. Search for singer Taylor Swift on Chrome and open her introduction page. |
| WhatsApp | 1. Send a hello message to "xxx" on WhatsApp. 
 2. Find the contact xxx and open the chat interface. | 1. Reply to unread messages on WhatsApp. 
 2. Reply to all the unread messages on WhatsApp. |
| *Multi-app* | | |
| - | 1. View the contacts and create a new note in Notes to record these contacts. 
 2. View the content of the note in Notes, then search for videos about it on TikTok. | 1. Check your unread messages in WhatsApp and search YouTube for videos related to that message. 
 2. Search the result for today's Thunder game, and then create a note in Notes to write a sport news for this result. |

Table 5: The applications and instructions used for the evaluation on English scenario, where the "xxx" represents the redacted information.

| App | Basic Instruction | Advanced Instruction |
|---|---|---|
| *System app* | | |
| 设置 | 1. 打开深色模式。
2. 切换系统主题。 | 1. 将系统的声音调为震动模式
2. 关闭通知栏的实时网速显示。 |
| 相机 | 1. 打开相机拍一张照片。

2. 打开相机录一个视频。 | 1.打开相机用长焦镜头拍一张照片，拍完之后查看该照片。
2. 打开相机用长焦镜头录一个视频，录完之后查看该视频。 |
| 通信 | 1. 给联系人xxx打电话。
2. 在信息中发送一条打招呼的短信给xxx。 | 1. 给电话号码为123的用户打电话。
2. 根据短信内容回复未读的短信。 |
| 备忘录 | 1. 新建一个备忘录，随便写点东西。

2. 新建一个备忘录，记录当前手机的电量和网络信号。 | 1. 在备忘录中删除现有的备忘录，然后新建一个备忘录，记录当前手机的电量和网络信号并保存。
2. 新建一个备忘录，写点东西后返回上一页面，再新建一个备忘录，记录当前手机的电量和网络信号。 |
| 系统应用 | 1. 在手机管家中优化手机。
2. 在应用市场安装"通义星辰"。 | 1. 在时钟中新建一个闹钟。
2. 在日历中新建一个日程。 |
| *External app* | | |
| 微信 | 1. 在微信中给xxx发送一个打招呼的消息。
2. 在微信中查找联系人"王君阳"并进入他的聊天界面。 | 1. 根据消息内容回复微信中的未读消息。
2. 根据消息内容回复微信中所有的未读消息。 |
| 小红书 | 1. 在小红书找一个帖子并评论相关内容。
2. 在小红书的消息中给王君阳发送一个打招呼的消息。 | 1. 根据消息内容回复小红书中的未读消息。
2. 在小红书中搜索一个机器学习相关的帖子并评论相关内容。 |
| 微博 | 1. 在微博发现页打开一条微博热搜的词条。
2. 在微博中搜索博主"雷军"并关注他 | 1. 根据帖子内容评论微博首页中的一个帖子。
2. 在微博中给"雷军"并发送一条私信，告诉他小米SU7真是太棒了。 |
| 抖音 | 1. 在抖音中上划刷出一个汽车相关的视频并点赞。

2. 在抖音中上划刷出一个汽车相关的视频并评论相关内容。 | 1. 在抖音中上划刷出一个汽车相关的视频并分享给其他用户。
2. 在抖音中搜索博主"雷军"并打开他的一条视频，然后评论相关内容。 |
| 哔哩哔哩 | 1. 在哔哩哔哩搜索"雷军"并关注他。
2. 在哔哩哔哩的首页找一个视频并评论相关内容。 | 1. 在哔哩哔哩找一个视频发一条弹幕。
2. 在哔哩哔哩找一个视频给出三连（点赞、投币、收藏）。 |
| *Multi-app* | | |
| - | 1. 查看今天的天气，然后退出并在备忘录中写一个穿衣指南。
2. 在微博的发现页查看一条热搜，然后退出并在备忘录中写一个该热搜的分析。 | 1. 在微博的发现页查看一条热搜，然后退出并在抖音中搜一个有关热搜的视频。
2. 查看微信中王君阳给你发来的消息，然后退出并在哔哩哔哩搜索一个与消息相关的视频。 |

Table 6: The applications and instructions used for the evaluation on non-English scenario, where the "xxx" represents the redacted information..

**System**
You are a helpful AI mobile phone operating assistant.

**User**
### Background ###
There is an user's instruction which is: {User's instruction}. You are a mobile phone operating assistant and are operating the user's mobile phone.

### Hint ###
There are hints to help you complete the user's instructions. The hints are as follow:
If you want to tap an icon of an app, use the action "Open app"

### Current operation ###
To complete the requirements of user's instruction, you have performed an operation. Your operation thought and action of this operation are as follows:
Operation thought: {Last operation thought}
Operation action: {Last operation}

### Response requirements ###
Now you need to combine all of the above to generate the "Completed contents". Completed contents is a general summary of the current contents that have been completed. You need to first focus on the requirements of user's instruction, and then summarize the contents that have been completed.

### Output format ###
Your output format is:
### Completed contents ###
Generated Completed contents. Don't output the purpose of any operation. Just summarize the contents that have been actually completed in the ### Current operation ###.
(Please use English to output)

Table 7: The prompt for the planning agent during the first operation.

**System**
You are a helpful AI mobile phone operating assistant.

**User**
### Background ###
There is an user's instruction which is: {User's instruction}. You are a mobile phone operating assistant and are operating the user's mobile phone.

### Hint ###
There are hints to help you complete the user's instructions. The hints are as follow:
If you want to tap an icon of an app, use the action "Open app"

### History operations ###
To complete the requirements of user's instruction, you have performed a series of operations. These operations are as follow:
Step-1: [Operation thought: {operation thought 1}; Operation action: {operation 1}]
Step-2: [Operation thought: {operation thought 2}; Operation action: {operation 2}]
......

### Progress thinking ###
After completing the history operations, you have the following thoughts about the progress of user's instruction completion:
Completed contents:
{Last "Completed contents"}

### Response requirements ###
Now you need to update the "Completed contents". Completed contents is a general summary of the current contents that have been completed based on the ### History operations ###.

### Output format ###
Your output format is:
### Completed contents ###
Updated Completed contents. Don't output the purpose of any operation. Just summarize the contents that have been actually completed in the ### History operations ###.

Table 8: The prompt for the planning agent during subsequent operations.

**System**

You are a helpful AI mobile phone operating assistant. You need to help me operate the phone to complete the user's instruction.

**User**

### Background ###
This image is a phone screenshot. Its width is {Lateral resolution} pixels and its height is {Vertical resolution} pixels. The user's instruction is: {User's instruction}.

### Screenshot information ###
In order to help you better perceive the content in this screenshot, we extract some information on the current screenshot through system files. This information consists of two parts: coordinates; content. The format of the coordinates is [x, y], x is the pixel from left to right and y is the pixel from top to bottom; the content is a text or an icon description respectively. The information is as follow:
(x1, y1); text or icon: text content or icon description
......

### Keyboard status ###
We extract the keyboard status of the current screenshot and it is whether the keyboard of the current screenshot is activated.
The keyboard status is as follow:
The keyboard has not been activated and you can't type. or The keyboard has been activated and you can type.

### Hint ###
There are hints to help you complete the user's instructions. The hints are as follow:
If you want to tap an icon of an app, use the action "Open app"

### History operations ###
Before reaching this page, some operations have been completed. You need to refer to the completed operations to decide the next operation. These operations are as follow:
Step-1: [Operation thought: {operation thought 1}; Operation action: {operation 1}]
......

### Progress ###
After completing the history operations, you have the following thoughts about the progress of user's instruction completion:
Completed contents:
{Task progress from planning agent}

### Response requirements ###
Now you need to combine all of the above to perform just one action on the current page. You must choose one of the six actions below:
Open app (app name): If the current page is desktop, you can use this action to open the app named "app name" on the desktop.
Tap (x, y): Tap the position (x, y) in current page.
Swipe (x1, y1), (x2, y2): Swipe from position (x1, y1) to position (x2, y2).
Unable to Type. You cannot use the action "Type" because the keyboard has not been activated. If you want to type, please first activate the keyboard by tapping on the input box on the screen. or Type (text): Type the "text" in the input box.
Home: Return to home page.
Stop: If you think all the requirements of user's instruction have been completed and no further operation is required, you can choose this action to terminate the operation process.

### Output format ###
Your output consists of the following three parts:
### Thought ###
Think about the requirements that have been completed in previous operations and the requirements that need to be completed in the next one operation.
### Action ###
You can only choose one from the six actions above. Make sure that the coordinates or text in the "()".
### Operation ###
Please generate a brief natural language description for the operation in Action based on your Thought.

Table 9: The prompt for the decision agent.

**System**

You are a helpful AI mobile phone operating assistant.

**User**

These images are two phone screenshots before and after an operation. Their widths are {Lateral resolution} pixels and their heights are {Vertical resolution} pixels.

In order to help you better perceive the content in this screenshot, we extract some information on the current screenshot through system files. The information consists of two parts, consisting of format: coordinates; content. The format of the coordinates is (x, y), x is the pixel from left to right and y is the pixel from top to bottom; the content is a text or an icon description respectively The keyboard status is whether the keyboard of the current page is activated.

### Before the current operation ###
Screenshot information:
(x1, y1); text or icon: text content or icon description
......
Keyboard status:
The keyboard has not been activated. or The keyboard has been activated.

### After the current operation ###
Screenshot information:
(x1, y1); text or icon: text content or icon description
......
Keyboard status:
The keyboard has not been activated. or The keyboard has been activated.

### Current operation ###
The user's instruction is: {User's instruction}. You also need to note the following requirements: If you want to tap an icon of an app, use the action "Open app". In the process of completing the requirements of instruction, an operation is performed on the phone. Below are the details of this operation:
Operation thought: {Last operation thought}
Operation action: {Last operation}

### Response requirements ###
Now you need to output the following content based on the screenshots before and after the current operation:
Whether the result of the "Operation action" meets your expectation of "Operation thought"?
A: The result of the "Operation action" meets my expectation of "Operation thought".
B: The "Operation action" results in a wrong page and I need to return to the previous page.
C: The "Operation action" produces no changes.

### Output format ###
Your output format is:
### Thought ###
Your thought about the question
### Answer ###
A or B or C

Table 10: The prompt for the reflection agent.

