# OpenReview forum: "Mobile-Agent-v2: Mobile Device Operation Assistant with Effective Navigation via Multi-Agent Collaboration"
_NeurIPS.cc/2024/Conference — NeurIPS 2024 poster_

### Official Review · Reviewer_4qiL · 2024-07-11

**Soundness:** 3
**Presentation:** 3
**Contribution:** 3
**Rating:** 7
**Confidence:** 4

**Summary:**

This paper proposes a multi-agent architecture for mobile device operation, Mobile-Agent-v2. Mobile-Agent-v2 includes three agents : planning agent, decision agent and reflection agent. To retain focus content, this paper designs a memory unit to record task-related focus content. The planning agent generates task progress based on the history operations, the reflection agent corrects erroneous operations and the decision agent outputs operations. Extensive experiments conducted on various operating systems, language environments, and applications shows that MobileAgent-v2 achieves significant performance improvements compared to single-agent architecture of Mobile-Agent.

**Strengths:**

1. This paper introduces a multi-agent architecture Mobile-Agent-v2, which can alleviate various navigating difficulties inherent in the single-agent framework for mobile device operation tasks.
2. This paper designs a memory unit and reflection agent, which can avoid the loss of focus content navigating and reflection capability.
3. Experimental results demonstrate that Mobile-Agent-v2 achieves significant performance improvements.

**Weaknesses:**

1. Insufficient baselines. More powerful baselines should be considered, such as AppAgent [1] and CogAgent [2].
2. This paper lack description of how the decision agent retrieves the memory unit. In addition,  the content or format of the memory unit storage is not included.

[1] AppAgent: Multimodal Agents as Smartphone Users. 2023.

[2] CogAgent: A Visual Language Model for GUI Agents. 2023.

**Questions:**

1. Can the Mobile-Agent-v2 be generalized to IOS systems?
2. How operations knowledge is injected into the model?

**Limitations:**

The authors describe limitations in the paper.

---

> ### Author Rebuttal · Authors · 2024-08-05
>
> ## **Question 1: Can Mobile-Agent-v2 be transferred to iOS or other platform?**
> ## **Response:**
>
> * Mobile-Agent adheres to a purely visual solution, making it universally applicable across platforms since Mobile-Agent-v1. Mobile-Agent-v2 continues this approach, allowing it to be transferred to any GUI-based system, such as tablets, PCs, TVs, and automotive systems.
>
> * Specifically, we have implemented it on PCs, facilitating operations such as "downloading papers in the browser" and "modifying font and formatting in Word." The relevant code will also be open-sourced on GitHub recently.
>
> ## **Question 2: How to implement knowledge injection.**
> ## **Response:**
>
> Operation knowledge is often a tutorial for a specific app. It is input into the decision model as part of the prompt:
> ```python
> if add_info != "":
>     prompt += f'''
> ### Hint ###
> There are hints to help you complete the user's instructions. The hints are as follows:
> {add_info}
> '''
> ```
> where "add_info" is the tutorial.
>
> For example, on TikTok, sharing a video requires clicking the fourth icon on the right. However, the LLM may not have this knowledge. In this case, knowledge can be injected: "The share icon is a white arrow pointing to the right on the right side of the screen." When Mobile-Agent-v2 reaches the step where it needs to click the share icon, it will complete the operation based on the injected knowledge. Since the use of knowledge is determined by the decision agent, other steps are not affected by the presence of the knowledge.
>
> ## **Question 3: Lack of other baselines such as CogAgent and AppAgent.**
> ## **Response:**
>
> **CogAgent**
>
> CogAgent is a QA model that does not possess the capability to perform concrete operations on real devices through tool invocation. Therefore, our dynamic evaluation framework is not applicable to such QA models.
>
> **AppAgent**
>
> First, AppAgent requires an additional exploration phase for each app, whereas Mobile-Agent-v2 can perform app operations without the exploration phase. Secondly, AppAgent relies on XML, restricting it to the Android platform. In contrast, Mobile-Agent-v2 uses a purely visual solution, making it platform-independent. Thus, comparing Mobile-Agent-v2 with AppAgent is not fair.
>
> Nevertheless, we manually evaluated the AppAgent and the results are shown in the below table. With the same base model, Mobile-Agent-v2 outperforms AppAgent.
>
> |Model|Success Rate|Completion Rate|
> |-|-|-|
> |AppAgent|66.7%|74.3%|
> |Mobile-Agent-v2|77.8%|82.1%|
>
> ## **Question 4: Memory unit storage format and retrieval method**
> ## **Response:**
>
> * Memory is stored as natural language descriptions of task-related content from historical screenshots. For example, in the task "Check the weather and write a dressing guide in notes," after opening the weather app, the agent needs to remember the weather information from the screenshot for subsequent use. Before exiting the weather app, the agent will add the weather information to the memory unit in plain text.
>
> * The stored memory is shared among all agents. When the agent needs to input the dressing guide, the decision agent will automatically retrieve the relevant information from the memory during inference.

---

> > ### Comment · Reviewer_4qiL · 2024-08-10
> > **Official Comment by Reviewer 4qiL**
> >
> > Thank you for taking the time and effort to address all questions.
> >
> > There are a few follow-up questions.
> >
> > a) Could you provide more details about the experiments comparing the performance of Mobile-Agent-v2 and AppAgent, such as the environment, setting, number of evaluation tasks, etc.
> >
> > b) How is the storage and retrieval costs of memory cells?

---

> > > ### Author Response · Authors · 2024-08-11
> > > **Response to the Official Comment by Reviewer 4qiL**
> > >
> > > Thank you for your response. Below is our response to the above two questions.
> > >
> > > ### **Question 1: Could you provide more details about the experiments?**
> > > ### **Response:**
> > > * Environment: We keep the evaluation environment consistent with the our paper, namely multi-platform (Android OS and Harmony OS), multi-app type (system app and third-party app), and multi-language (English and non-English). All evaluation tasks start from the mobile phone's desktop, and the app used in the instructions is ensured to exist on the desktop.
> > > * Setting: We used AppAgent's official open source project for evaluation. During the evaluation, we only conducted an exploration phase, that is, running the official code “learn.py”. All other settings, including hyper-parameters and MLLM selection, were defaulted using the official code.
> > > * Number: We hope that AppAgent can be consistent with the paper in terms of apps and evaluation tasks. However, since AppAgent relies on XML, some pages involved in instructions cannot obtain XML files. We found that this may be because the app page has permissions or the page is dynamic (such as video), which makes the XML unable to be obtained through ADB. We also removed these tasks from the evaluation results of AppAgent and Mobile-Agent-v2. In the end, a total of 63 instructions were used for evaluation.
> > >
> > > ### **Question2: How is the storage and retrieval costs of memory cells?**
> > > ### **Response:**
> > > * The storage of memory unit depends on the maximum input length of MLLMs. For example, the most advanced MLLM GPT-4o can support an input length of 128K. This means that the memory unit can definitely hold the entire app's tutorials. These tutorials can come from the official instructions of the app, manually written operating experience, or from key information recorded during the historical tasks of Mobile-Agent-v2.
> > > * Since memory retrieval is done at the decision stage by decision agent, we cannot directly quantify the retrieval efficiency. However, we have experimentally found that, while keeping the output length unchanged, every 1k tokens added to the memory unit will increase the time it takes the decision agent to make a decision by 200ms to 1s. This means that even if the memory length reaches 3k tokens (the length of a general app tutorial), the decision stage will only increase the time by 8% (the average decision time increased from 21s to 22.8s.).

---

> > > > ### Comment · Reviewer_4qiL · 2024-08-11
> > > > **Official Comment by Reviewer 4qiL**
> > > >
> > > > Thanks for your reply, things are clear now. I hope that these experiments details and results can be reflected in a future version of the paper.

---

### Official Review · Reviewer_cAKE · 2024-07-13

**Soundness:** 2
**Presentation:** 2
**Contribution:** 2
**Rating:** 5
**Confidence:** 4

**Summary:**

This paper proposes mobile-agent-v2, a multi-agent framework for mobile device operation. The framework includes a planning agent, decision agent, reflection agent and memory unit. The multi-agent framework exhibits advantages in reducing errors in long-horizon tasks and achieves better results than single-agent framework on the evaluations designed in the aper.

**Strengths:**

1. Multi-agent framework for mobile device operation is novel in the literature, improving the overall performance of single agent framework, with the effectiveness of each component verified.

**Weaknesses:**

1 The paper lacks a comparison of the evaluation benchmark with previous works such as AppAgent and works mentioned in Sec. 2.2. This makes it difficult to evaluate the overall effectiveness of the framework in comparison with previous works.
2. The evaluation benchmark is relatively small and simple, and how they are evaluated is not very clear. Does the evaluation relies on human evaluator to check the success of each component in the trajectory?

**Questions:**

1. See weakness, the evaluation details needs further clarification.

**Limitations:**

1. This paper relies on commercial models such as GPT-4 and GPT-4V to complete the tasks. The cost of completing a task is uncaculated or discussed in the paper, but is expected to be more cost than the single agent framework.
2. The efficiency and accuracy trade-off. After introducing mutli-agent framework, the inference time should be much slower.

--------
After the rebuttal, the reviewer think the heavy reliance on commercial models and the time and token cost should be discussed in the revision.

---

> ### Author Rebuttal · Authors · 2024-08-05
>
> ## **Question 1: Lack of other baselines such as CogAgent and AppAgent.**
> ## **Response:**
>
> **CogAgent**
>
> CogAgent is a QA model that does not possess the capability to perform concrete operations on real devices through tool invocation. Therefore, our dynamic evaluation framework is not applicable to such QA models.
>
> **AppAgent**
>
> First, AppAgent requires an additional exploration phase for each app, whereas Mobile-Agent-v2 can perform app operations without the exploration phase. Secondly, AppAgent relies on XML, restricting it to the Android platform. In contrast, Mobile-Agent-v2 uses a purely visual solution, making it platform-independent. Thus, comparing Mobile-Agent-v2 with AppAgent is not fair.
>
> Nevertheless, we manually evaluated the AppAgent and the results are shown in the below table. With the same base model, Mobile-Agent-v2 outperforms AppAgent.
>
> |Model|Success Rate|Completion Rate|
> |-|-|-|
> |AppAgent|66.7%|74.3%|
> |Mobile-Agent-v2|77.8%|82.1%|
>
> ## **Question 2: Small and simple evaluation set with unclear evaluation method.**
> ## **Response:**
>
> * Mobile-Agent-v2 uses dynamic evaluation, requiring the agent to directly invoke mobile operation tools and connect to a real device for evaluation. This method is more complex and challenging compared to static app screenshots used in existing work, which only evaluate single-step operations.
>
> * Mobile-Agent-v2's dynamic evaluation involves 20 apps, including both system and third-party apps, across various operating systems and languages. Each app has 4 tasks, with an average of 7 steps per task. Each step undergoes three evaluations: planning, decision, and reflection, totaling 1500+ evaluations. This evaluation scale is the largest among works using dynamic evaluation, covering the most apps and having the most comprehensive task coverage.
>
> * We also received endorsements from Reviewer **c2St** and **Xdcr** for our evaluation method: **"The paper includes a detailed evaluation across different operating systems, language environments, and applications, providing robust evidence of the system's effectiveness."** and **"The experimental results on real-world mobile apps look promising."** We also recognize the community's lack of a general benchmark based on dynamic evaluation. We are currently working on creating this benchmark and providing a rich set of operation tools to encourage more models to participate in dynamic evaluation.
>
> ## **Question 3: Token cost of usage.**
> ## **Response:**
> Assuming an average of 7 steps per task, Mobile-Agent-v2, using a multi-agent architecture, consumes a relatively fixed number of tokens per step. In contrast, a single-agent architecture requires inputting a long sequence of operation history and screenshots with each call, leading to increased token consumption as the number of task steps rises.
>
> |Architecture|Token per Step|Total Token|
> |-|-|-|
> |Mobile-Agent (single-agent)|(1.3k) * steps + 0.4k|43.4k|
> |Mobile-Agent-v2 (multi-agent)|4.4k|30.8k|
>
> ## **Question 4: Efficiency reduction in multi-agent systems.**
> ## **Response:**
>
> * For time overhead, although multiple agents work serialize, there are still phases to parallel. The tables below compare the operation times for the Mobile-Agent single-agent framework and the Mobile-Agent-v2 multi-agent framework. "Screenshot" represents obtaining a screenshot and corresponding image processing from the device, while "Tool Call" represents the use of visual perception tools. It is evident that the increased operation time of the multi-agent framework is acceptable under parallel design.
>
> * For memory overhead, Mobile-Agent-v2 calls the large model via API method, so there is no additional memory overhead.
>
> **Mobile-Agent (Single-Agent Framework)**
> |Phase|Preparation|Planning|Decision|Operation|Reflection|Total|
> |-|-|-|-|-|-|-|
> |Task|Screenshot|-|Decision|Tool Call, Grounding|-||
> |Time|2s||20s|12s|-|**34s**||
>
> **Mobile-Agent-v2 (Multi-Agent Framework)**
> |Phase|Preparation|Planning|Decision|Operation|Reflection|Total|
> |-|-|-|-|-|-|-|
> |Task|Parallel with Planning|Screenshot, Tool Call, Planning, Reflection|Decision, Memory|-| Parallel with Planning||
> |Time|-|18s|21s| -| -|**39s**||

---

> > ### Comment · Reviewer_cAKE · 2024-08-13
> > **A few further questions**
> >
> > Thank the authors for the detailed response.
> > However, the reviewer is still confused about whether the successful judgment is done by humans or GPT-4V. If it is by GPT-4V, to what degree is the assessment reliable?
> > As mentioned in lines 215-216 of the main paper, there are 88 instructions tested in total, with only 8 instructions for multi-app operations, what is the relationship between the so-called 1500+ evaluation with the 88 instructions?
> >
> > Mobile Agent-v2 seems to increase the overall time to finish the task in comparison with Mobile agent-v1, not to mention it is impractical to use GPT-4V for mobile devices. Further, the time calculation does not consider the information transmission time through the web or cable in real-world applications when using GPT-4V.

---

> > > ### Author Response · Authors · 2024-08-13
> > > **Response to the Official Comment by Reviewer cAKE**
> > >
> > > ### **Question 1: How to determine whether the operation is successful or not?**
> > > ### **Response:**
> > > Due to the lack of dynamic evaluation benchmarks in the mobile field, all our success or failure judgments are made through manual evaluation.
> > >
> > > ### **Question 2: What is the relationship between the 1500+ evaluation with the 88 instructions?**
> > > ### **Response:**
> > > The instructions used in the Mobile-Agent-v2 evaluation require multiple steps to complete, with an average of 7 steps per instruction. For each operation, we manually evaluated the accuracy of planning, decision-making, and reflection. Therefore, there are about 88 x 7 x 3 = 1848 evaluations in total.
> > >
> > > ### **Question 3: Mobile Agent-v2 seems to increase the overall time to finish the task in comparison with Mobile agent-v1.**
> > > ### **Response:**
> > > * First although Mobile-Agent-v2 consumes about 15% more time, it achieves an operation success rate of more than 30%. Both are based on the same MLLM, and Mobile-Agent-v2 can significantly improve its operational capabilities by virtue of its architectural advantages.
> > > * The main reason that currently limits the operation time of Mobile-Agent-v2 is the inference speed of GPT-4. This is not a flaw in the framework itself. As the inference speed of MLLMs continues to accelerate and more advanced inference acceleration methods are used, or the use of local MLLMs, the inference speed will be further shortened. This will also significantly reduce the time-consuming gap between single-agent architecture and multi-agent architecture, while allowing the operation speed to be close to that of humans.
> > > * It is worth noting that although the operation speed of Mobile-Agent is not as fast as that of humans, there are still many practical application scenarios where such operation delays can be accepted. In addition, thanks to the parallel logic between agents, Mobile-Agent is also the fastest architecture that can currently achieve real machine operation. We are currently working on using local models to replace GPT-4 and have achieved initial results. Currently, the operation time per operation can be as low as 10 seconds.
> > >
> > > ### **Question 4: The time calculation.**
> > > ### **Response:**
> > > The time we calculate is the time of the complete operation, which includes the network delay and the communication consumption on the pipeline. If Mobile-Agent-v2 is used in a real device or app, it will not increase the operation delay.

---

> > > > ### Comment · Reviewer_cAKE · 2024-08-13
> > > > **Thanks for the author response**
> > > >
> > > > Thanks for the clarification.
> > > > The reviewer tends to raise the score to 5 since most of the questions are resolved.
> > > > However, the concern about it heavy reliance on GPT-4V and the time cost still exists. It is not very straightforward to think about the scenarios that accept the slow time and GPT-4V token cost, which should be added in the discussion or limitation section in the next revision.

---

### Official Review · Reviewer_Xdcr · 2024-07-15

**Soundness:** 2
**Presentation:** 3
**Contribution:** 2
**Rating:** 5
**Confidence:** 4

**Summary:**

This paper introduces an agentic framework, Mobile-Agent-v2, designed to address the challenges of planning and sequential function/tool invocation for Large Language Models (LLMs) in mobile operation scenarios. Mobile-Agent-v2 comprises three agents: a planner, an actioner, and a reflector, which jointly enhance the performance of LLMs in mobile operation tasks. Experimental results demonstrate the advantage of Mobile-Agent-v2 over the single-agent framework: Model-Agent, across various real-world applications.

**Strengths:**

- It is well-written and the structure is clear.
- Its studied problem of how to design agentic workflow for LLM to improve planning and sequential tool calling is beneficial to the community.
- The experimental results on real-world mobile apps look promising.

**Weaknesses:**

- The technical contribution seems limited. Its proposed framework of using planner, actioner and reflector for agentic workflow is a common practice.
- The evaluation misses some important ablation studies.

**Questions:**

This paper proposes an agentic workflow, comprising a planner, actioner, and reflector, to enhance the planning and sequential tool-calling capabilities of Large Language Models (LLMs) in mobile operations. The motivation behind this approach is sound, and the experimental results appear promising. However, I have several concerns primarily regarding the technical novelty and the insights derived from the experiments.

- The method of decomposing the responsibilities of a single agent into multiple components—planner, actioner, and reflector—is not novel. This approach has been previously proposed by [1] and is widely used in other contexts. Therefore, it is essential to clarify the key challenges and new insights associated with applying this framework to mobile operation scenarios. Without such clarification, it appears that the study merely leverages an existing method and applies it to a specific scenario without significant innovation.

- The experimental results across various mobile applications are promising. However, the sequence length and type are critical factors that warrant detailed discussion. While some analysis is provided, the paper lacks comprehensive results regarding task completion accuracy when sequence length and type vary. For instance, it would be beneficial to understand, across different apps, which types of operations are more successfully completed with different tool sequences. These ablation studies are necessary to offer deeper insights.

---

> ### Author Rebuttal · Authors · 2024-08-05
>
> ## **Question 1: Similar multi-agent architectures have been applied in other scenarios.**
> ## **Response:**
>
> Mobile-Agent-v2 is the first work to use a multi-agent architecture in the mobile domain. The required capabilities of each agent, the tasks they perform, the interaction between agents, and their independent units are significantly different from existing work. Below are the challenges faced by Multi-modal Large Language Models (MLLMs) in the mobile domain and the solutions provided by our framework:
>
> 1. **Input type and sequence length:** In the mobile domain, MLLMs' operation decisions face the problem of multi-image long sequences and interleaved text and images input format, which limits MLLMs' decision accuracy. We propose a planning agent to maintain the stability of input sequence length and format in the single image. This input can improve the context learning of MLLMs.
> 2. **Operation grounding:** In the mobile domain, MLLMs need to generate decisions and their locations. Currently, mainstream MLLMs (even GPT-4) lack grounding capabilities. To address this, we included a visual perception tool to assist decision-making. This purely visual solution overcomes the limitations of the operating platform and offers greater versatility.
> 3. **Error gandling in long sequences:** In the mobile domain, long sequence operation tasks inevitably lead to errors during intermediate operations. When errors occur, MLLMs often struggle to reflect effectively and correct mistakes. We introduced a reflection agent that uses the MLLMs' multi-image understanding capabilities and contextual learning to judge the accuracy of operations and correct errors.
> 4. **Following historical screen information:** In the mobile domain, following and navigating history screen information is difficult due to input length limitations and interleaved text and images. We proposed using an independent memory unit to store this information, significantly enhancing the agent's ability to navigate key information.
>
> Regarding the design of the Mobile-Agent-v2 framework, we note positive feedback from other reviewers, such as **c2St** and **cAKE**, who endorsed our approach: **"The introduction of a multi-agent system to handle different aspects of mobile device operation is a novel approach"**; **"Multi-agent framework for mobile device operation is novel in the literature"**. This confirms the novelty of applying a multi-agent framework in the mobile domain.
>
> Moreover, Mobile-Agent-v2 is a practical framework. Our code, released in the open-source community, has received widespread acclaim, garnered thousands of stars, and has been applied in various real-world scenarios. This practical application stands out among the many works on multi-agent frameworks.
>
> ## **Question 2: The relationship between sequence length and operation type requires further study.**
> ## **Response:**
>
> We have compiled statistics on the relationship between operation sequence length and operation accuracy in the table below.
>
> |Sequence Length|Open|Click|Swipe|Type|Back|Home|Stop|
> |-|-|-|-|-|-|-|-|
> |[1, 4)|86.4%|91.5%|100%|75.0%|-|100%|100%|
> |[4, 7)|100%|81.3%|100%|60.0%|100%|100%|88.4%|
> |[7, )|-|80.8%|-|75.0%|-|-| 86.2%|
>
> Here are the conclusions:
> 1. **Minimal impact of sequence length in simple operations:** Operations that do not require precise coordinates or parameters, such as "Swipe", "Back", and "Home", are minimally affected by sequence length.
> 2. **Success rate variation in complex operations:** For click and stop operations, the success rate is higher at the beginning of the task than in the middle or later stages. This is because early-stage operations usually involve primary pages with better guidance, while advanced pages require the agent to have more robust operational knowledge. However, with the multi-agent architecture, there is no significant decline in the middle and later stages despite the sequence length.

---

> > ### Author Response · Authors · 2024-08-14
> > **Response to the Official Comment by Reviewer Xdcr**
> >
> > Thank you very much for your feedback and the time you dedicated to reviewing our paper. We greatly appreciate your insights and are pleased to hear that the clarifications we provided were helpful in addressing your concerns. If you have any further questions or suggestions, please do not hesitate to reach out. We would be happy to discuss them with you.

---

> ### Comment · Reviewer_Xdcr · 2024-08-14
>
> Thank the authors for their rebuttals. I will raise my score to 5. However, the results on the multi-modal scenarios suggest that the performance of the proposed agentic framework can be satisfactory only when it has been applied on GPT-4V. This may limit the generalization of the proposed agent frameworks. In the next version of this paper, I would like to see whether the gap between applying Mobile-agent-V2 on GPT-4V and on other open-source models, especially on small models (less than 7B) can be narrowed down. This is because small models are more practical choice when deploying LLMs on the mobile device.

---

### Official Review · Reviewer_kbfL · 2024-07-16

**Soundness:** 3
**Presentation:** 3
**Contribution:** 2
**Rating:** 4
**Confidence:** 3

**Summary:**

In this paper, the authors present Moble-Agent-v2, a  multi-agent architectruure deisgned to assist mobile device operation. Mobile-Agent-v2 comprises three agents: planning agent, decision agent, and reflection agent. Frist, planning agent summarizes the task progress based on the operation hsistories. Then, the decision agent generates next operation to be executed. Finally, the reflextion agent classifies whether the excuted operation as correct, errorneous, or ineffective. For evaluation, the authors conduct experiments on real mobile devices in various mobile applications.

**Strengths:**

- The application of multi-modal LLMs in mobile operation is intriguing
- The framework that uses multiple agents demnostrates superior perfornamce compared to the single-agent framework
- The paper is easy to follow and evaluation is conducted on the real mobile devices

**Weaknesses:**

- The primary concern is the novelty of the paper. While using multiple agents for mobile operations appears novel, the complexity of tasks used in the evaluation suggests that the implementation of each component within Mobile-Agent-V2 is relatively simple. It would be benifical if the authors evalute the proposed framework in diverse user scenarios (refer to the limitation section).
- It appears that the authors assume that LLMs possess the knowledge about mobile applications. For example, if the user instruction is "turn on dark mode," how does the LLM determine which setting (general setting, display setting) the agent should execute? Because, sometimes even for the human need the trial and error to execute certain task.  Without grounding the available execution in the given context, the proposed method might require multiple trial and error until it successfully completes the task.
- Regarding to the upper question, it would be better if the authors provide how many interactions are required to complete a  task compared to the optimal execution.
- It seems to be neither erroneous nor ineffective operations are recorded in the operation history. If this is the case, isn’t there a potentional risk that the decision agent might repeat the same operation? Woudn’t it be better to use these operation histories to prevent the LLM from repeating erroneous or ineffective executions?
- There seems some potential risks associated with inaccuracies in every agents within the proposed framework, especially since it utilizes LLMs. For instance, the planning agent might inaccurately summarize operation histories, the decision agent could misunderstand visual inputs, or it might generate incorrect executions. Can Mobile-Agent-v2 automatically identify which component is responsible for an error? or could the authors provide more statistical information about this?

**Questions:**

- Can the authors provide more information about knowledge injection? It would be helpful if the authors included examples to illustrate this process.

**Limitations:**

- Including some discussion on the scope limitations of Mobile-Agent-v2 would be beneficial. For instance, what happens if the required application is missing—can Mobile-Agent-v2 suggest an alternative method? Additionally, for ambiguous instructions, is Mobile-Agent-v2 capable of resolving these by seeking clarification from the user?

---

> ### Author Rebuttal · Authors · 2024-08-05
>
> ## **Question 1: Lack of evaluation in special scenarios, such as unachievable or ambiguous tasks.**
> ## **Response:**
>
> * For tasks that cannot be directly completed through the given instructions, Mobile-Agent-v2 can still attempt to complete them. Due to the high flexibility of Mobile-Agent-v2's operation space, it can simulate almost any operation on a mobile device. Therefore, even if the device does not have the conditions to complete the task, Mobile-Agent-v2 can try to create the conditions through mobile operations. For example, if the task is to open "Facebook" but the app is not installed on the device, Mobile-Agent-v2 will open the app store, search for "Facebook," and click to install it. Then it will return to the home screen and open the app.
> * For ambiguous instructions that require explicit user input, Mobile-Agent-v2 can resolve them by expanding the operation space. Mobile-Agent-v2 supports custom operations within the operation space. For example, an operation "seek help" can be added with the description: "Use this operation when you have multiple operation paths to get clearer instructions from the user." When multiple choices can fulfill the instruction's requirement, Mobile-Agent-v2 can determine that the current state is ambiguous and proactively invoke the corresponding operation to obtain user guidance.
> * Due to inherent biases in Multi-modal Large Language Models (MLLMs), some operations with strict triggering conditions may be difficult for the MLLMs to use. We are also working on improving the MLLMs‘ contextual learning and operation invocation abilities through alignment training and reinforcement learning.
>
> ## **Question 2: How will the agent operate if the agent lacks operation knowledge?**
> ## **Response:**
>
> * Mobile-Agent-v2 can leverage the operation knowledge within the MLLMs and the multi-agent cooperation to complete complex tasks. Existing mainstream MLLMs, such as GPT-4, Gemini, and Qwen-VL, have operational experience with many apps. Even if the MLLMs lacks knowledge of certain apps, the decision agent can infer possible operations based on page content and the output of the planning agent. Even if errors occur, they can be corrected by the reflection agent. Additionally, operational capabilities can be acquired through training, such as Apple's Ferret-UI.
>
> * We propose knowledge injection to explore whether external operation knowledge can compensate for the agent's operational deficiencies. If a task is too difficult for the MLLMs' internal knowledge, knowledge injection can generate tutorials. Details on knowledge injection will be addressed in the next question.
>
> * If knowledge injection is not used, the agent can acquire operation knowledge through self-exploration. The agent will perform possible operations and use the reflection agent to observe the results, determining whether the operation was correct. The knowledge gained from exploration is then added to an external knowledge base as input for the decision agent. This way, the agent will not need to explore when facing the same task again. We selected samples from the test set with exploration processes and recorded the steps required to complete the task in the table below, where "KI" represents knowledge injection. We can see that the efficiency of the additional exploration process is acceptable.
> |w/o KI|w/ KI|
> |-|-|
> |8.6|6.4|
>
> ## **Question 3: How to implement knowledge injection?**
> ## **Response:**
>
> Operation knowledge is often a tutorial for a specific app. It is input into the decision model as part of the prompt:
> ```python
> if add_info != "":
>     prompt += f'''
> ### Hint ###
> There are hints to help you complete the user's instructions. The hints are as follows:
> {add_info}
> '''
> ```
> where "add_info" is the tutorial.
>
> For example, on TikTok, sharing a video requires clicking the fourth icon on the right. However, the LLM may not have this knowledge. In this case, knowledge can be injected: "The share icon is a white arrow pointing to the right on the right side of the screen." When Mobile-Agent-v2 reaches the step where it needs to click the share icon, it will complete the operation based on the injected knowledge. Since the use of knowledge is determined by the decision agent, other steps are not affected by the presence of the knowledge.
>
> ## **Question 4: No record of incorrect operations in operation history.**
> ## **Response:**
> To simplify the structure of the prompt, we have omitted the situation when an incorrect operation occurs. We save the incorrect operations in another part of the decision agent's inputs:
> ```python
> if error_flag:
>     prompt += f'''
> ### Last operation ###
> You previously wanted to perform the operation \"{last_summary}\" on this page and executed the Action \"{last_action}\". But you find that this operation does not meet your expectation. You need to reflect and revise your operation this time.'''
> ```
>
> ## **Question 5: Can the framework automatically attribute errors to specific modules or provide error statistics?**
> ## **Response:**
> * Thank you for your question. This is an important capability for UI interaction agents. However, since the modules within the framework are interdependent, it is challenging to attribute errors to a specific module solely through global reflection after each operation.
>
> * We manually analyzed the causes of errors in Mobile-Agent-v2 on the evaluation set. The results show that errors can occur at any stage.
> |Planning Agent|Visual Tool Results|Decision Agent|
> |-|-|-|
> |12|10|13|
>
> Therefore, to achieve automated error detection, independent reflections need to be designed for each module. For improvement, we will add reflection on the previous module's results while generating results for each module, preventing errors from propagating between modules. This will not increase latency due to the parallel design. Moreover, errors can be intercepted at the point of occurrence. This will be a key direction for our future work.

---

> ### Comment · Reviewer_kbfL · 2024-08-12
>
> I thank authors for their detailed responses to my questions.
>
> While each element of Mobile-Agent-v2 may not be novel in itself, its significance lies in configuring the agentic workflow through multiple agents within the mobile application domain. Therefore, I believe that this paper should be evaluated from an end-user perspective. From this viewpoint, it is essential for the framework to automatically identify and resolve the errors, as highlighted in question 5. Additionally, minimizing the exploration, as discussed in question 2, is crucial for enhancing the user experience. Moreover, regarding question 1, the proposed framework might handle ambiguous tasks, but there is a lack of experimental proof.
>
> As these details crucial to enhancing the user experience are missing in Mobile-Agent-v2, I will maintain my current score. I hope these aspects can be addressed in future version of the paper.

---

> > ### Author Response · Authors · 2024-08-12
> > **Response to the Official Comment by Reviewer kbfL**
> >
> > Thank you for your continued evaluation and thoughtful feedback. We appreciate your recognition of the significance of configuring an agentic workflow within the mobile application domain.
> >
> > We understand your concern regarding the practical utility of Mobile-Agent-v2 compared to existing methods. It is worth emphasizing that our approach has demonstrated a substantial improvement in task completion rates and a significant reduction in errors during navigation tasks. These results directly translate to an enhanced user experience, particularly in complex mobile operation scenarios. The ability of Mobile-Agent-v2 to efficiently manage long sequences and interleaved data in real-world applications is a notable advantage over other related methods.
> >
> > The framework's design is specifically optimized to minimize user intervention and exploration during mobile operations, which is critical for maintaining a smooth and intuitive user experience.
> >
> > * **Automatically attribute errors.** We agree that automatic error resolution and handling ambiguous tasks are critical aspects. This study is focused on establishing the foundational architecture of Mobile-Agent-v2, specifically addressing the challenges of task progress navigation and content focus in mobile operations.
> >
> > * **end-user evaluation.** The area of mobile GUI agent is still at very early stage. The current time and resource constraints make end-user evaluations challenging. Our primary focus at this stage has been on establishing the core architecture and demonstrating its effectiveness through technical benchmarks, which we believe are essential first steps before moving on to broader end-user evaluations.
> >
> > We believe that Mobile-Agent-v2 is well-positioned to significantly advance mobile operation tasks, and we are committed to improving the framework based on your valuable suggestions.

---

### Official Review · Reviewer_c2St · 2024-07-17

**Soundness:** 3
**Presentation:** 3
**Contribution:** 3
**Rating:** 6
**Confidence:** 4

**Summary:**

The paper titled "Mobile-Agent-v2: Mobile Device Operation Assistant with Effective Navigation via Multi-Agent Collaboration" presents a multi-agent architecture designed to address the challenges of mobile device operation tasks, specifically focusing on task progress navigation and focus content navigation. The proposed system includes three agents: a planning agent, a decision agent, and a reflection agent. The architecture aims to improve task completion rates and operational efficiency compared to existing single-agent architectures.

**Strengths:**

**Innovative Multi-Agent Architecture:** The introduction of a multi-agent system to handle different aspects of mobile device operation is a novel approach. It effectively distributes the workload among specialized agents, which likely contributes to improved performance.

**Comprehensive Experimental Evaluation:** The paper includes a detailed evaluation across different operating systems, language environments, and applications, providing robust evidence of the system's effectiveness.

**Significant Performance Improvements:** Experimental results show that Mobile-Agent-v2 achieves over a 30% improvement in task completion compared to a single-agent architecture, demonstrating the practical benefits of the proposed system.

**Weaknesses:**

**Limited Discussion on Scalability:** There is insufficient discussion on the scalability of the proposed system. The paper does not address potential challenges when scaling the architecture to more complex tasks or larger datasets.

**Potential Overhead of Multi-Agent Coordination:** While the multi-agent approach shows improved performance, the paper does not discuss the potential computational overhead and complexity introduced by coordinating multiple agents, which could be a significant drawback in resource-constrained environments.

**Questions:**

- How does the proposed system handle the increased computational overhead associated with running multiple agents concurrently on resource-constrained mobile devices?
- Can the system be extended to support more complex multi-app operations that require intricate coordination between different agents?
- How does the reflection agent determine the appropriate corrective measures for erroneous operations, and what is the success rate of these corrections in practice?

**Limitations:**

see weakness

---

> ### Author Rebuttal · Authors · 2024-08-05
>
> ## **Question 1: Overhead of multi-agent.**
> ## **Response:**
> * For time overhead, although multiple agents work serialize, there are still phases to parallel. The tables below compare the operation times for the Mobile-Agent single-agent framework and the Mobile-Agent-v2 multi-agent framework. "Screenshot" represents obtaining a screenshot and corresponding image processing from the device, while "Tool Call" represents the use of visual perception tools. It is evident that the increased operation time of the multi-agent framework is acceptable under parallel design.
>
> * For memory overhead, Mobile-Agent-v2 calls the large model via API method, so there is no additional memory overhead.
>
> **Mobile-Agent (Single-Agent Framework)**
> |Phase|Preparation|Planning|Decision|Operation|Reflection|Total|
> |-|-|-|-|-|-|-|
> |Task|Screenshot|-|Decision|Tool Call, Grounding|-||
> |Time|2s||20s|12s|-|**34s**||
>
> **Mobile-Agent-v2 (Multi-Agent Framework)**
> |Phase|Preparation|Planning|Decision|Operation|Reflection|Total|
> |-|-|-|-|-|-|-|
> |Task|Parallel with Planning|Screenshot, Tool Call, Planning, Reflection|Decision, Memory|-| Parallel with Planning||
> |Time|-|18s|21s| -| -|**39s**||
>
> ## **Question 2: Can Mobile-Agent-v2 be extended to more complex multi-app scenarios?**
> ## **Response:**
> * Yes, Mobile-Agent-v2 can be extended to more complex multi-app scenarios. Mobile-Agent-v2 is a purely visual general framework that is not restricted by app type or operating system, allowing it to be freely extended to any scenario. Mobile-Agent-v2 inherently possesses the capability to plan and make decisions in complex scenarios. Benefiting from the planning agent not being affected by long sequence and image-text interleaved inputs, the decision agent can ensure decision accuracy. Additionally, the memory unit can store key information from previously opened apps' screenshot for use in subsequent operations.
>
> * For extremely complex multi-app scenarios, Mobile-Agent-v2 also supports custom extensions to the operation space. By simply adding the functional description of the operation to the operation space, Mobile-Agent-v2 can perform the operation when needed. For example, if the user needs to use the Mobile-Agent-v2 to crawl the screenshot of the mobile app in batches, the user can add the "screenshot" operation, and request the Mobile-Agent-v2 in the instruction to use the operation to intercept the screen after completing the specified operation task. This extension can effectively improve accuracy without affecting operation efficiency.
>
> ## **Question 3: How does the reflection agent determine if an operation is correct and the success rate of error correction?**
> ## **Response:**
> * The Multi-modal Large Language Model (MLLM) itself has multi-image understanding capabilities, which supports multi-image input reflection. While the decision agent outputs an operation, it also outputs the operation intent. The internal contextual reasoning ability of the MLLM can be used to determine whether the operation result meets expectations. Additionally, the MLLM has some operation knowledge, so even if the decision agent's operation intent is inaccurate, it can still make a judgment based on the user's task.
>
> * We have compiled statistics on the success rate and average steps required for the reflection agent's error correction in the table below, where "Reflection SR," "Correction SR," and "Average Steps" respectively represent the success rate of reflection, the success rate of error correction, and the average steps for error correction. The results show that over **60%** of operational errors can be successfully corrected, with the cost being less than **2** steps.
> |Reflection SR|Correction SR|Average Steps|
> |-|-|-|
> |94%|62%|1.63|

---

### Author Rebuttal · Authors · 2024-08-05

We sincerely appreciate the reviewers for their valuable and constructive feedback, which will be pivotal in enhancing the quality of our work.

We are encouraged by the following reviewers' perceptions:
* Innovative and interesting multi-agent architecture in the mobile domain (c2St, kbfL, cAKE).
* Comprehensive experiments are promising (c2St, Xdcr).
* Significant performance improvements (c2St, kbfL, cAKE, 4qiL).
* Well-written and easy to follow (kbfL, Xdcr).

We appreciate the valuable suggestions and questions raised by the reviewers regarding Mobile-Agent-v2. These insights are significantly important for refining our work and guiding future research. We have diligently addressed all concerns and questions from the reviewers in individual responses. Below, we highlight some of the key issues or frequently asked questions raised by the reviewers.

## **Question 1: Overhead of multi-agent.**
## **Response:**
* For time overhead, although multiple agents work serialize, there are still phases to parallel. The tables below compare the operation times for the Mobile-Agent single-agent framework and the Mobile-Agent-v2 multi-agent framework. "Screenshot" represents obtaining a screenshot and corresponding image processing from the device, while "Tool Call" represents the use of visual perception tools. It is evident that the increased operation time of the multi-agent framework is acceptable under parallel design.

* For memory overhead, Mobile-Agent-v2 calls the large model via API method, so there is no additional memory overhead.

**Mobile-Agent (Single-Agent Framework)**
|Phase|Preparation|Planning|Decision|Operation|Reflection|Total|
|-|-|-|-|-|-|-|
|Task|Screenshot|-|Decision|Tool Call, Grounding|-||
|Time|2s||20s|12s|-|**34s**||

**Mobile-Agent-v2 (Multi-Agent Framework)**
|Phase|Preparation|Planning|Decision|Operation|Reflection|Total|
|-|-|-|-|-|-|-|
|Task|Parallel with Planning|Screenshot, Tool Call, Planning, Reflection|Decision, Memory|-| Parallel with Planning||
|Time|-|18s|21s| -| -|**39s**||

## **Question 2: How to implement knowledge injection?**
## **Response:**

Operation knowledge is often a tutorial for a specific app. It is input into the decision model as part of the prompt:
```python
if add_info != "":
    prompt += f'''
### Hint ###
There are hints to help you complete the user's instructions. The hints are as follows:
{add_info}
'''
```
where "add_info" is the tutorial.

For example, on TikTok, sharing a video requires clicking the fourth icon on the right. However, the LLM may not have this knowledge. In this case, knowledge can be injected: "The share icon is a white arrow pointing to the right on the right side of the screen." When Mobile-Agent-v2 reaches the step where it needs to click the share icon, it will complete the operation based on the injected knowledge. Since the use of knowledge is determined by the decision agent, other steps are not affected by the presence of the knowledge.

## **Question 3: Similar multi-agent architectures have been applied in other scenarios.**
## **Response:**

Mobile-Agent-v2 is the first work to use a multi-agent architecture in the mobile domain. The required capabilities of each agent, the tasks they perform, the interaction between agents, and their independent units are significantly different from existing work. Below are the challenges faced by Multi-modal Large Language Models (MLLMs) in the mobile domain and the solutions provided by our framework:

1. **Input type and sequence length:** In the mobile domain, MLLMs' operation decisions face the problem of multi-image long sequences and interleaved text and images input format, which limits MLLMs' decision accuracy. We propose a planning agent to maintain the stability of input sequence length and format in the single image. This input can improve the context learning of MLLMs.
2. **Operation grounding:** In the mobile domain, MLLMs need to generate decisions and their locations. Currently, mainstream MLLMs (even GPT-4) lack grounding capabilities. To address this, we included a visual perception tool to assist decision-making. This purely visual solution overcomes the limitations of the operating platform and offers greater versatility.
3. **Error gandling in long sequences:** In the mobile domain, long sequence operation tasks inevitably lead to errors during intermediate operations. When errors occur, MLLMs often struggle to reflect effectively and correct mistakes. We introduced a reflection agent that uses the MLLMs' multi-image understanding capabilities and contextual learning to judge the accuracy of operations and correct errors.
4. **Following historical screen information:** In the mobile domain, following and navigating history screen information is difficult due to input length limitations and interleaved text and images. We proposed using an independent memory unit to store this information, significantly enhancing the agent's ability to navigate key information.

## **Question 4: Lack of other baselines such as CogAgent and AppAgent.**
## **Response:**

**CogAgent**

CogAgent is a QA model that does not possess the capability to perform concrete operations on real devices through tool invocation. Therefore, our dynamic evaluation framework is not applicable to such QA models.

**AppAgent**

First, AppAgent requires an additional exploration phase for each app, whereas Mobile-Agent-v2 can perform app operations without the exploration phase. Secondly, AppAgent relies on XML, restricting it to the Android platform. In contrast, Mobile-Agent-v2 uses a purely visual solution, making it platform-independent. Thus, comparing Mobile-Agent-v2 with AppAgent is not fair.

Nevertheless, we manually evaluated the AppAgent and the results are shown in the below table. With the same base model, Mobile-Agent-v2 outperforms AppAgent.

|Model|Success Rate|Completion Rate|
|-|-|-|
|AppAgent|66.7%|74.3%|
|Mobile-Agent-v2|77.8%|82.1%|

---

> ### Author Response · Authors · 2024-08-14
>
> Thank you very much for your feedback and the time you dedicated to reviewing our paper. We greatly appreciate your insights and are pleased to hear that the clarifications we provided were helpful in addressing your concerns. If you have any further questions or suggestions, please do not hesitate to reach out. We would be happy to discuss them with you.

---

### Decision · Program_Chairs · 2024-09-25

**Decision:**

Accept (poster)

**Comment:**

This paper describes Mobile-Agent-v2, an MLLM-based multi-agent architecture designed to assist with mobile device operations, incorporating the multi-agent collaboration with three agents: a planning agent, a decision agent, and a reflection agent.

The paper was reviewed by five reviewers with four inclined to suggest for acceptance and one not. The majority of reviewers highlighted the strengths, such that (1) the proposed fmobile-agent-v2 framework is an interesting contribution to the mobile domain, (2) the paper is clearly written and easy-to-follow, and (3) the experiments with real-world mobile apps demonstrate promising and significant performance. One reviewer commented several concerns about limited operational knowledge of existing MLLMs and error-correcting capabilities of the agent, which the authors argued are beyond the scope of current research work. However, since this research addresses the challenges in practical mobile device operation tasks, those concerns could be discussed further in the final version.

Considering the strengths above and the novel approach toward agentic workflow in the mobile domain, this paper is recommended for acceptance.
The authors are encouraged to utilize their constructive and detailed discussion with reviewers towards further raising the quality and impact of their work.